# Sensitive dependence of trajectories on tracer seeding positions - coherent structures in German Bight backward drift simulations

Ulrich Callies[1]

[1]Institute of Coastal Research, Helmholtz-Zentrum Geesthacht, Max-Planck-Str. 1, 21502 Geesthacht, Germany

**Correspondence:** Ulrich Callies (ulrich.callies@hzg.de)

**Abstract.** Backward drift simulations can aid the interpretation of in situ monitoring data. In some cases, however, trajectories are very sensitive to even small changes of the tracer release position. A corresponding spread of backward simulations implies attraction in the forward passage of time and hence uncertainty about the probed water body's origin. This study examines surface drift simulations in the German Bight (North Sea). Lines across which drift behaviour changes non-smoothly are obtained as ridges in the fields of the finite-time Lyapunov exponent (FTLE), a parameter used in dynamical systems theory to identify Lagrangian coherent structures (LCS). Results closely resemble those obtained considering two-particle relative dispersion. It is argued that simulated FTLE fields might be used in support of the interpretation of monitoring data, indicating when simulations of backward trajectories are unreliable because of their high sensitivity to tracer seeding positions.

## 1 Introduction

In the German Bight area (North Sea) a comprehensive monitoring network is operated, including the Marine Environmental Monitoring Network in the North Sea (MARNET), the Coastal Observing System for the North and Arctic Seas (COSYNA) and other stations. Details on the type of data being collected can be found in Baschek et al. (2017). Stanev et al. (2016) discuss issues related to modelling and data assimilation with spatiotemporal optimal interpolation. Multivariate statistical methods can also be used for optimizing the design of observational arrays (e.g. Chen et al., 2016; Kim and Hwang, 2020). When it comes to the analysis of specific data, however, a merely statistical description of spatial connectivity falls short of what can be achieved if hydrodynamic current fields from either models or remote sensing are available.

Backward tracer trajectories seeded at monitoring stations provide insight into the background of water bodies that were probed (e.g. d'Ovidio et al., 2015; Lucas et al., 2016; Teeling et al., 2012, supplement, movie S1). They help distinguish between temporal and spatial variability, i.e. local changes and advection from somewhere else. Backtracking water bodies from hypothetical monitoring stations in the vicinity of Helgoland, Callies et al. (2011, their Fig. 3) provide an example of how quasi-chaotic mixing in two-dimensional barotropic simulations transforms initially regular into quite contorted structures. Also in nature, forward trajectories of drifters released pairwise may separate quite fast (e.g. Callies et al., 2019; Meyerjürgens

et al., 2020). Therefore, a key question is how reliable backward drift analyses can be and how the numerical analysis of a water body's recent history should be designed. In addition to well known random dispersion there exist also flow patterns that affect separation of simulated backward trajectories more systematically (Haller, 2015). The present study focuses on this latter aspect, coherent structures shaping separation of simulated backward trajectories.

A statistical measure of particle spreading is relative dispersion, the mean square particle distance, as function of time. LaCasce (2008) reviews how this parameter relates to the energy spectrum of a turbulent flow. Relative dispersion is called local when particle separation is dominated by small eddies with a typical scale that compares with particle separation. By contrast, it is termed non-local if particle separation is dominated by eddies much larger than particle separation. In the latter case, characterized by a steep energy spectrum, particle separation is expected to grow exponentially. Such very high sensitivity to initial particle positions implies what in dynamical systems theory is called chaotic advection (Wiggins, 2005).

Dynamical systems theory aims at a description of the kinematics of turbulent mixing. The approach is based on flow maps that describe particle advection over some time interval, according to Haller (2015) "*thereby mimicking experimental flow visualization by tracers*". This technique has widely been applied for analysing the microstructure of chaotic mixing processes in two dimensions (e.g. Pierrehumbert and Yang, 1993), describing how chaotic advection may transform initially small disks of fluid into complex filamentary structures. Trying to improve the sometimes vague definitions of such structures, Haller and Yuan (2000) introduced the framework of Lagrangian coherent structures (LCS). Their method seeks to identify material lines that function as only weakly permeable barriers for water body transport, attracting or repelling neighboured trajectories. Peacock and Haller (2013) provide a nice overview of the topic.

Attracting LCSs, in dynamical systems theory also called unstable because of a fast stretching of particles along them (according to Harrison and Glatzmaier (2010) an unfortunate historical definition), have been used for optimizing drifter deployments in field studies. Poje et al. (2002) proposed drifter deployment into attracting LCSs to ensure fast dispersal based on near-exponential material stretching, which lets drifters explore regions of high kinetic energy. Molcard et al. (2006) used this approach for assimilating drifter velocities into an ocean general circulation model. Different from these studies, Shadden et al. (2009) focus on repelling LCSs. Seeding drifters in a less localized way, Shadden et al. try to make drifters stay as long as possible in a specific region delineated by transport barriers. They exemplify that a LCS's robustness might enable extrapolation of its separatrix function even beyond the time horizon of detailed operational hydrodynamic predictions (e.g. three days). Combining SeaWiFS ocean-colour data with altimetry-derived surface currents in the Brazil-Malvinas confluence zone, d'Ovidio et al. (2010) found that stirring by mesoscale currents can play an important role in structuring phytoplankton communities and even create what they call fluid dynamical niches, sharply delimited by LCSs. Hernández-Carrasco et al. (2018) study this topic at the submesoscale, using currents observed with High-Frequency Radar (HFR) in coastal waters. According to Scales et al. (2018) attracting LCSs can also be targeted by fisheries, led by lines of drifting foam or debris.

Conducting backward simulations, the present study proposes the use of LCSs as indicators of a possibly sensitive dependence of measurements on where and when exactly they were taken. The analysis is based on offline drift simulations using German Bight surface layer currents obtained from archived output of the operational 3D baroclinic model BSHcmod, run operationally by the German Federal Maritime and Hydrographic Agency (BSH). The study aims for an assessment of the situ-

ations at specific times of interest rather than for a generic characterization or classification of given locations. Highly variable transport paths in the German Bight area for the most part arise from residual currents driven by changing wind conditions (Schrum, 1997; Callies et al., 2017a). Establishing a simple interrelationship between winds and overall finite-time transports is hardly possible as tracer trajectories may aggregate the effects of very different winds. Detailed numerical simulations, however, properly integrate such variable hydrodynamic transports during a specific observation period.

Hadjighasem et al. (2017) compare twelve candidate approaches that could be used for the identification of LCSs. Among those, calculation of finite-time Lyapunov exponents (FTLE) is one of the most common methods. The FTLE is closely related to the finite-size Lyapunov exponent (FSLE), originally introduced by Aurell et al. (1996, 1997) and used in experiments for diagnosing scale dependent separation rates between drifter pairs (LaCasce and Ohlmann, 2003; Sansón et al., 2017). Karrasch and Haller (2013), however, report some limitations for FSLE in LCS detection, suggesting that an approach based on FTLE distributions may be more reliable. The FTLE fields are independent of an observer's reference frame (Haller, 2015), representing the rate at which neighbouring tracers separate according to the largest eigenvalue of the so-called Cauchy-Green strain tensor.

Building on work by Haller (2001), Shadden et al. (2005) define LCS in terms of ridges in the FTLE field. In two dimensions, the LCSs are material lines transported with the flow that, in good approximation, act as transport barriers. Discussing some counterexamples in which substantial material flows across a FTLE ridge occurred, Haller (2011) developed a more sophisticated variational approach. Farazmand and Haller (2012) presented a corresponding numerical algorithm for two-dimensional applications. Recently Tian et al. (2019) applied a variational method to identify the outer bounds of the Kuroshio current system. Here, the analysis will adhere to conventional FTLE fields.

The paper is organized as follows: Section 2 first describes the study area and how Lagrangian drift simulations were performed based on pre-calculated hydrodynamic surface current fields. It follows a short compilation of the definitions of the FTLE, stretch, Lagrangian divergence and statistical measures of dispersion. Presenting a couple of structures that emerged under different wind conditions, Section 3 then provides an overview of the type of LCSs that can be found in the German Bight area. Example trajectories substantiate the relevance of FTLE ridges as material separatrices. A general discussion and a short summary conclude the paper.

## 2 Material and methods

### 2.1 Study area

The North Sea is a semi-enclosed shelf sea that connects to the north-eastern Atlantic at its northern boundary and through the English Channel at its southwest (Sündermann and Pohlmann, 2011). Strong tidal forcing occurs as a co-oscillation triggered by Atlantic tidal waves. This study focusses on the German Bight, the shallow south-eastern part of the North Sea (here: east of 6.5°E and south of 56°N) with water depths of mainly 20-50 m (see Fig. 1), adjoining the Dutch, the German and the Danish coasts (Becker et al., 1992). In the German Bight, a mean cyclonic North Sea circulation corresponds with residual currents from the southwest to the north. Superimposed to this mean circulation, a strong weather driven variability occurs on short

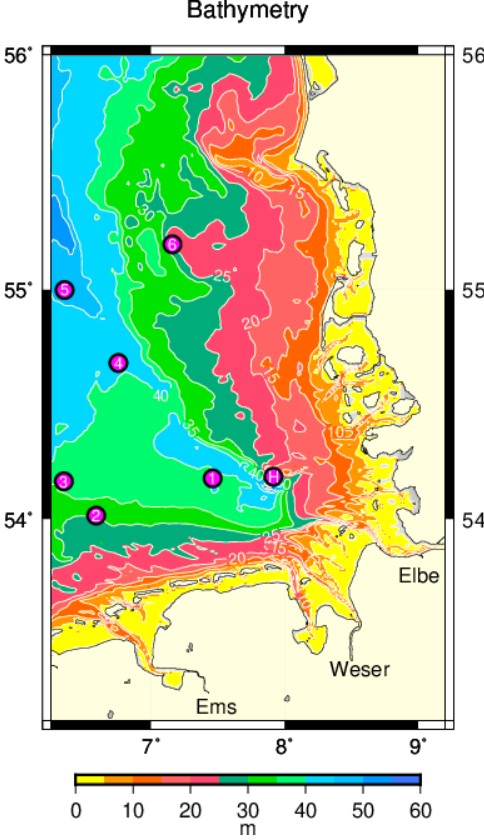

**Figure 1.** German Bight bathymetry. Magenta coloured circles indicate locations of six stations of the MARNET monitoring network (labels 1-6) and of the island of Helgoland (label H).

time scales (Schrum, 1997; Callies et al., 2017a). A fresh water plume emerging from the Elbe River and, to a minor extent, also the Weser and the Ems rivers can be observed as a permanent feature. Transient eddies and meanders depend on bottom topography, baroclinic instabilities and wind effects.

The most important topographic feature is the old Elbe Glacial Valley, opening from today's Elbe estuary towards the northwest into the open North Sea (see Fig. 1). Becker et al. (1992) summarize different types of fronts (river plume, thermal and upwelling fronts) that occur in the German Bight. Frontal structures depend on season but vary also on a short term basis (Budéus, 1989; Schrum, 1997). In the warm season, strong stratification occurs at water depths greater than approximately 30 m, mainly in the Elbe Glacial Valley. A baroclinic tidal mixing front (James, 1984; Holt and Umlauf, 2008) separates this

region from the well-mixed more shallow coastal water, where stratification is prevented by strong tidal mixing (Krause et al., 1986).

## 2.2   Hydrodynamic fields

Surface layer currents used for offline drift simulations (see Section 2.3) as well as temperature fields were taken from archived BSHcmod model output. BSHcmod is a three-dimensional baroclinic circulation model, formulated using geographical coor-

105 dinates and a flexible vertical resolution that allows for weakly inclined coordinate surfaces (Dick et al., 2008) of up to 36 layers. The model is run operationally by the Federal Maritime and Hydrographic Agency (BSH) since many years, providing the basis for different oceanographic services including search-and-rescue applications. It covers both the North Sea and the Baltic Sea and is two-way nested with approximately 900 m resolution in the German Bight area (Fig. 1) and approximately 5 km in the open North Sea (Dick et al., 2001). The domain of the present analysis roughly agrees with the region of 900 m

resolution in German North Sea coastal waters.

     In BSHcmod, advection and diffusion are calculated using a flux-corrected transport scheme. The hydrostatic and the Boussinesq approximations are applied. The Smagorinsky scheme (Smagorinsky, 1963) is used for the parameterization of horizontal viscosity. For an inclusion of wind stress, the parametrization by Smith and Banke (1975) is used. Stokes drift remains disregarded in the operational model output. The model's atmospheric forcing is provided by the regional model COSMO-EU

(Consortium for Small-Scale Modelling; Schulz and Schättler (2014)), run by the German Meteorological Service (Deutscher Wetterdienst – DWD). The Swedish Meteorological and Hydrological Institute (SMHI) and the Federal Institute of Hydrology (BfG) provide runoff data for all major rivers that flow into the North Sea.

     In the process of archiving, BSHcmod hydrodynamic fields with originally higher vertical resolution were re-gridded. Conserving transport rates, this was done in such a way that the archived surface currents used in this study are representative for

the uppermost 5 metres of the water column. With drift simulations based on these currents plus a leeway of 0.6 % of winds in 10 m height, Callies et al. (2017b, 2019) reproduced observed drifter trajectories reasonably well.

## 2.3   Lagrangian drift simulations

Offline drift simulations based on BSHcmod surface layer currents (archived on a 15 min basis) were performed using the Lagrangian transport program PELETS-2D (Callies et al., 2011). Originally, the PELETS toolbox developed at Helmholtz-

125 Zentrum Geesthacht was designed for its use with hydrodynamic currents on unstructured triangular grids. Current fields provided on a regular grid (like those from BSHcmod) must be preprocessed, splitting each rectangular grid cell into two triangles, a transformation of grid topology that does not affect the information content of hydrodynamic fields.

All simulations in this study were produced using the fourth-order Cash Karp method (Press et al., 1992) that belongs to the Runge Kutta family of solvers. A simple Euler forward scheme, however, used in other PELETS applications (e.g. Callies et al., 2011, 2017b, 2019) gave very similar results. The maximum time step is set to 15 min. Velocities are updated earlier every time that a tracer particle moves to another triangular grid cell.

## 2.4 Finite-time Lyapunov exponents (FTLE)

Definition of the FTLE is based on a consideration of Lagrangian flow motions. A flow map $\Phi$ maps particle locations $\boldsymbol{x}_0$, where particles were seeded at time $t_0$, onto their destinations $\boldsymbol{x}$ at later time $t = t_0 + \tau$:

$$\Phi_{t_0}^{\tau}(\boldsymbol{x_0}) = \boldsymbol{x}(t_0 + \tau; t_0, \boldsymbol{x}_0) \, . \tag{1}$$

The following deformation gradient describes material deformation,

$$\nabla \Phi_{t_0}^{\tau}(\boldsymbol{x_0}) = \begin{pmatrix} \frac{\partial x}{\partial x_0} & \frac{\partial x}{\partial y_0} \\ \frac{\partial y}{\partial x_0} & \frac{\partial y}{\partial y_0} \end{pmatrix} \, , \tag{2}$$

where $\boldsymbol{x} = (x, y)$. The deformation's Jacobian provides the ratio of the area $A$ of a deformed quadrangle to the area $A_0$ of an infinitesimal square it has its origin in. In case of a finite size initial square and a non-linear flow, this ratio refers to a quadrangle that approximates the emerging distorted patch. Similarly, a linear map sends an initially small circle to an ellipse. The lengths of the image area's semi-axes are given by the deformation gradient's two singular values $\mu_1$, $\mu_2$, whose product equals the Jacobian determinant:

$$\frac{A}{A_0} = \det\left(\nabla \Phi_{t_0}^{\tau}\right) = \mu_1 \mu_2 \, . \tag{3}$$

From the above deformation gradient, one obtains the following Cauchy-Green strain or deformation tensor (e.g. Shadden et al., 2005; Haller, 2015):

$$C(\tau; t_0, \boldsymbol{x}_0) = \left[\nabla \Phi_{t_0}^{\tau}(\boldsymbol{x_0})\right]^T \nabla \Phi_{t_0}^{\tau}(\boldsymbol{x_0}) \, . \tag{4}$$

This two-dimensional, symmetric and positive-definite tensor has two eigenvalues $\lambda_1 = \mu_1^2$ and $\lambda_2 = \mu_2^2$. Definition of the finite-time Lyapunov exponent (FTLE) is based on the tensor's larger eigenvalue $\lambda_1$:

$$\text{FTLE}\left(\tau; t_0, \boldsymbol{x}_0\right) = \frac{1}{|\tau|} \log \sqrt{\lambda_1} = \frac{1}{|\tau|} \log(\mu_1) \, . \tag{5}$$

The absolute value of integration time $\tau$ is used because integration of particle drift can be conducted either forward or backward in time. The geometric interpretation of the FTLE refers to the maximum separation rate of neighbouring particles (Haller, 2015).

To compute FTLE fields numerically, a regular Cartesian grid of tracers was released, initial locations with 1 km resolution covering the German Bight area east of 6.5°E and south of 56°N (165 vortices in the longitudinal and 310 vortices in the latitudinal direction). The corresponding 51150 trajectories were integrated 250 hours back in time ($\tau$ = -250 h). To avoid the

computational burden of four close-by auxiliary trajectories, finite-differencing in Eq. (2) was performed based on neighbouring trajectories seeded on the regular FTLE grid.

In view of the limited vertical resolution of archived BSHcmod currents, values of the deformation gradient (Eq. (2)) were tagged as missing each time at least one of the four tracer trajectories needed for its discrete calculation encountered a water depth of less than 5 m sometime in the course of its integration. Resulting gaps in the fields of FTLE and related quantities change with variable atmospheric forcing. As BSHcmod covers the whole North Sea, no specific treatment is needed for particles that cross the open boundaries of the FTLE grid.

## 2.5 Distinction between divergence and stretch

An incompressible two-dimensional flow field preserves the area of a Lagrangian patch during arbitrary deformations. Here, however, two-dimensional surface currents being used were extracted from three-dimensional hydrodynamic fields that allow for vertical exchange of water masses. Huntley et al. (2015) developed a concept that splits FTLE values into contributions that come from area-preserving stretching and deformation on the one hand and area changes on the other. Given the singular values $\mu_1 \geq \mu_2$ of the deformation gradient $\nabla \Phi_{t_0}^{\tau}(\boldsymbol{x_0})$, Huntley et al. (2015) introduce the following stretch rate $\Sigma$:

$$\Sigma = \frac{1}{|\tau|} \log\left(\frac{\mu_1}{\mu_2}\right) . \tag{6}$$

In addition, they introduce the following dilation rate $\Delta$ that describes the transformation of a infinitesimal Lagrangian patch's initial area $A_0$ to an area $A$ after integration time $\tau$,

$$\Delta = \frac{1}{|\tau|} \log\left(\frac{A}{A_0}\right) = \frac{1}{|\tau|} \log\left(\mu_1\mu_2\right) , \tag{7}$$

where Eq. (3) has been used. From Eq. (5) it follows that the separation rate represented by the FTLE can be decomposed in terms of the above two components:

$$\text{FTLE} = \frac{\Delta + \Sigma}{2} . \tag{8}$$

Dilation rate $\Delta$ can be shown to equal the average Eulerian horizontal divergences experienced by a fluid parcel along its pathway (Huntley et al., 2015; Duran et al., 2018, supplement). From the material derivative

$$\frac{dA}{dt} = A\nabla_H \cdot \boldsymbol{v} , \tag{9}$$

it follows that

$$A(\tau) = A_0 \exp\left(\int_{t_0}^{t_0+\tau} \nabla_H \cdot \boldsymbol{v}(t', \boldsymbol{x}(t'; t_0, \boldsymbol{x}_0))dt'\right) . \tag{10}$$

Hernández-Carrasco et al. (2018) refer to the patch area's change rate, derived from past Eulerian divergences, as the Finite-Domain Lagrangian Divergence (FDLD),

$$\text{FDLD}\left(\tau; t_0, \boldsymbol{x}_0\right) = \frac{1}{|\tau|} \log\left(\frac{A}{A_0}\right) = \frac{1}{|\tau|} \int_{t_0}^{t_0+\tau} \nabla_H \cdot \boldsymbol{v}(t', \boldsymbol{x}(t'; t_0, \boldsymbol{x}_0))dt' . \tag{11}$$

They demonstrate its potential for supporting the interpretation of satellite based observations of surface chlorophyll *a* patches.

Analytically, the FDLD from Eq. (11) equals dilation rate $\Delta$ from Eq. (7). In the present study, however, the Eulerian divergences needed for the numerical evaluation of Eq. (11) were computed based on auxiliary points introduced at a 250 m distance. Corresponding velocities were obtained by linear interpolation. As a result of this refinement, truncation errors due to numerical discretization differ from those of analyses on the basic 1 km FTLE grid.

### 2.6 Absolute and relative dispersion

Absolute and relative dispersion are statistical measures for analysing Lagrangian data. Generally, absolute dispersion is defined as the second moment of the single particle displacement PDF, i.e. the variance of particle displacements relative to their starting position. This measure must not be confused with cloud variance (LaCasce, 2008). Ensemble averaging could be performed with respect to either different locations or different realizations at some fixed location. Here, following Haller and Yuan (2000), the simpler density of absolute dispersion is considered, describing just one single particle's squared displacement from its release point:

$$a^2(\tau; t_0, \boldsymbol{x}_0) = \mid \boldsymbol{x}(t_0 + \tau; t_0, \boldsymbol{x}_0) - \boldsymbol{x}_0) \mid^2 \tag{12}$$

By contrast, relative dispersion describes the mean square separation of particle pairs with nearby initial release points. Relative dispersion at each node of the FTLE grid will be calculated combining the information from four particle pairs,

$$D^2(\tau; t_0, \boldsymbol{x}_0) = \frac{1}{4} \sum_{i=1}^{4} \mid \boldsymbol{x}(t_0 + \tau; t_0, \boldsymbol{x}_0) - \boldsymbol{x}(t_0 + \tau; t_0, \boldsymbol{x}_0 + \delta \boldsymbol{x}_i) \mid^2 \tag{13}$$

where $\delta \boldsymbol{x}_i$ denotes the distance vector between neighbouring nodes. For a comparison with FTLE and FDLD fields, the logarithm of absolute and relative dispersion is a reasonable choice. Exponential growth of pair separations indicates the presence of Lagrangian chaos dynamical systems theory deals with (Wiggins, 2005).

## 3 Results

A couple of examples will be given, intended to illustrate the occurrence and time variability of Lagrangian structures in German Bight surface currents. All analyses being shown will be based on a backward integration time of -250 h. This specific choice has, however, no crucial impact on the structures being shown. None of the sometimes sharp structures observed are persistent, occurrence and specific patterns depend on the past evolution of atmospheric forcing. For this reason, all figures contain wind roses that summarize the wind conditions during the previous 250 hours.

### 3.1 FTLE and Lagrangian divergence

Fig. 2a shows the FTLE field for simulations initialized on 12 June 2015 (13:00 UTC) and extending over 250 hours backward in time. The graph leaves blank all locations from where trajectories reached regions with water depths below 5 m. At the time

Fig. 2 refers to, the most prominent feature in the FTLE field is an extended south-north running ridge of high FTLE values. Further west, a less pronounced parallel second ridge occurs which, however, tends to be split into three or four segments. Other more local and sometimes also weaker filamentary structures can be recognized. Intended to illustrate the physical relevance of the central FTLE-ridge, Fig. 2a includes three groups of four 250 h backward trajectories, initialized in the wider neighbourhood of stations 1, 4 and 6 of the MARNET monitoring network[1]. To facilitate orientation and comparison, locations of the six MARNET stations and the island of Helgoland (station H) will be indicated in all further figures.

The experiment shown refers to a situation with calm conditions during the last three days, weak winds blowing from the north/northeast under the influence of a high pressure system centred further west. Vectors in the top right corner of Fig. 2a show simulated 10 m wind directions near MARNET station 4 during the past 250 hours at ten hourly intervals. Wind speeds are represented by a colour code. The wind vector at the time of the plot is edged in red, those during the last 50 hours are edged in black. Strong winds (> 17m/s) from south/southwest occurred on the second and third of June, i.e. at the end of the 250 hours backward integration period. Another period with enhanced wind speeds (< 10 m/s) occurred about 4-6 days before the time of the FTLE field being shown. At that time, winds from the southwest/west changed to winds from roughly northern directions.

Both the directional changes and the higher drift speeds at the end of the integration back in time can also be recognized from the example drift trajectories displayed in Fig. 2a. Two pairs of hypothetical in situ observation points (indicated by small circles, green and blue) were located on either side of the central FTLE ridge near MARNET stations 1 and 6. Trajectory end points are indicated by small diamonds. Simulations show a clear separation of those backward trajectories that emerge from different sides of the FTLE ridge. By contrast, trajectories started on the same side of the ridge (having same colour) remain close to each other. The examples illustrate how even close by in situ observations may encounter water bodies with a much different history. Fig. 2a shows also a complementary experiment, in which release points near MARNET station 4 are now all located within an area of low FTLE values. In this case all trajectories stay close together or even further converge.

---

[1]Station names: Deutsche Bucht (1), FINO1 (2), Ems (3), Nordseeboje III (4), Nordseeboje II (5), FINO3 (6)

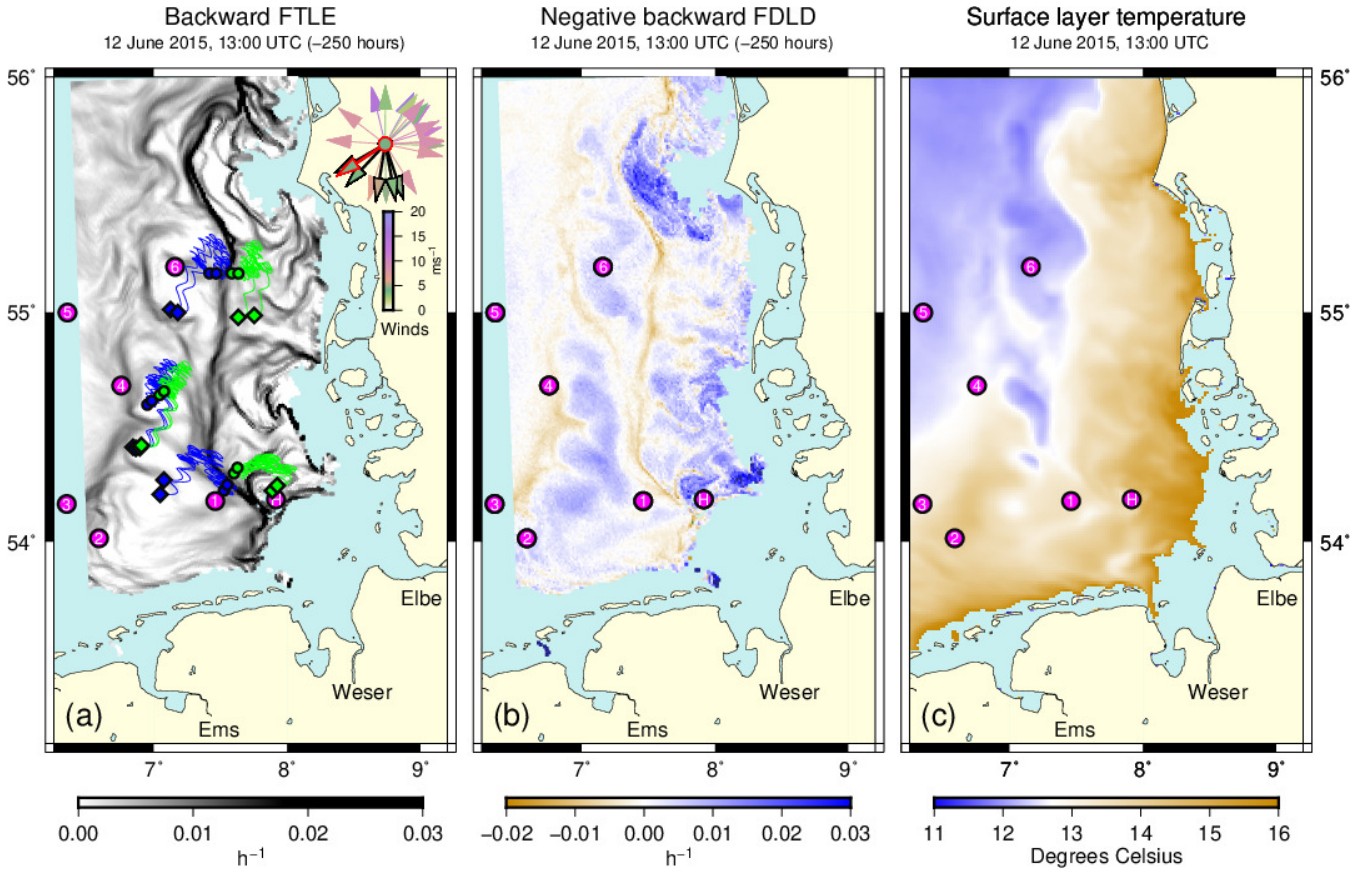

**Figure 2. (a)** FTLE field analysed for 12 June 2015 (13:00), evaluating Eq. (5) based on trajectories calculated 250 hours backward in time. The scale was chosen to well visualize ridges of large values, negative values of the logarithmic FTLE have been plotted as if they were zero. Locations from where trajectories encountered a water depth of less than 5 m sometime in the course of their integration were disregarded. Example backward trajectories are shown, using different colours (green/blue) for better distinction. Trajectory release points are indicated by circles, end points by diamonds. Labelled circles (magenta) indicate locations of six stations of the MARNET monitoring network (1-6) and of the island of Helgoland (H). Vectors in the top right corner show past wind directions at ten hourly intervals, referring to simulations for the location of MARNET station 4. Different wind speeds are represented by different colours. The wind vector at the time of the plot is edged in red, those during the last 50 hours are edged in black. **(b)** Negative Lagrangian divergences (FDLD) calculated from Eq. (11). Few large positive values, exceeding the range covered by the colour map, are plotted in dark blue. **(c)** Simulated mean temperatures in the uppermost 5 meters. To focus on open sea conditions, temperatures higher than $16°$C are not shown.

Separation in backward time means confluence in ordinary forward time. Therefore, the negative backward Lagrangian divergence FDLD (see Eq. (11)) shown in Fig. 2b is to be read in agreement with the usual passage of time. Fig. 2b reveals a striking structural similarity with Fig. 2a. Water parcels located on backward FTLE ridges have predominantly experienced converging surface currents along their pathway during the last 250 hours. Between the ridges, there are wider regions with particles the history of which was dominated by divergent Eulerian currents.

Studying the Agulhas current in the southwest Indian Ocean, van Sebille et al. (2018, their Fig. 3) found structures in fields of sea surface temperature (SST) that agreed with LCSs derived from geostrophic currents. For the present example, Fig. 2c shows a south-north oriented zone of relatively cool surface layer water, located in between narrow bands of higher temperature that tend to coincide with the FTLE ridges (Fig. 2a) and zones of convergence (Fig. 2b). The belt is made up by a couple of patches that bear a structural resemblance to patches of positive divergence in Fig. 2b, suggesting that some features of the temperature distribution in Fig. 2c can indeed be explained in terms of up- and downwelling simulated in the model. Meyerjürgens et al. (2020) found reduced relative dispersion for experimental drifters released in the vicinity of a tidal mixing front, indicating horizontal attraction in this region.

## 3.2 Influences of bathymetry and wind conditions

Fig. 3b shows a backward FTLE field that is even more clearly structured than the one in Fig. 2a, including also pronounced west-east oriented divides. Note that FTLE ridges in the western part of the domain closely follow the bathymetric feature of the old Elbe Glacial Valley (see Fig. 1). Again, the origins of example tracers, estimated by backward trajectories with close by release points (green/blue) on either side of FTLE ridges, vastly differ. Particularly large differences occur for the most northern and the most southern trajectries.

Similar to the example shown in Fig. 2, calm atmospheric conditions prevailed also for a couple of days preceding the time of Fig. 3b (26 March 2018, 18:00). Very strong easterly winds, however, persisted for a couple of days towards the end of the 250 h backward integration period (roughly 16-18 March). Nearly constant easterly wind directions for the last 50 hours can be distinguished from the wind rose in Fig. 3a, showing the FTLE field eight days earlier. Trajectories of North Sea drifters observed under these rare conditions (due to low temperatures in the UK called the 'Beast from the East') have been discussed by Stanev et al. (2019, see Fig. 3a therein). Some FTLE ridges that emerge according to Fig. 3a are aligned parallel with the easterly winds and seem to correspond with or to transform into sharp FTLE ridges in Fig. 3b under the subsequent much calmer conditions.

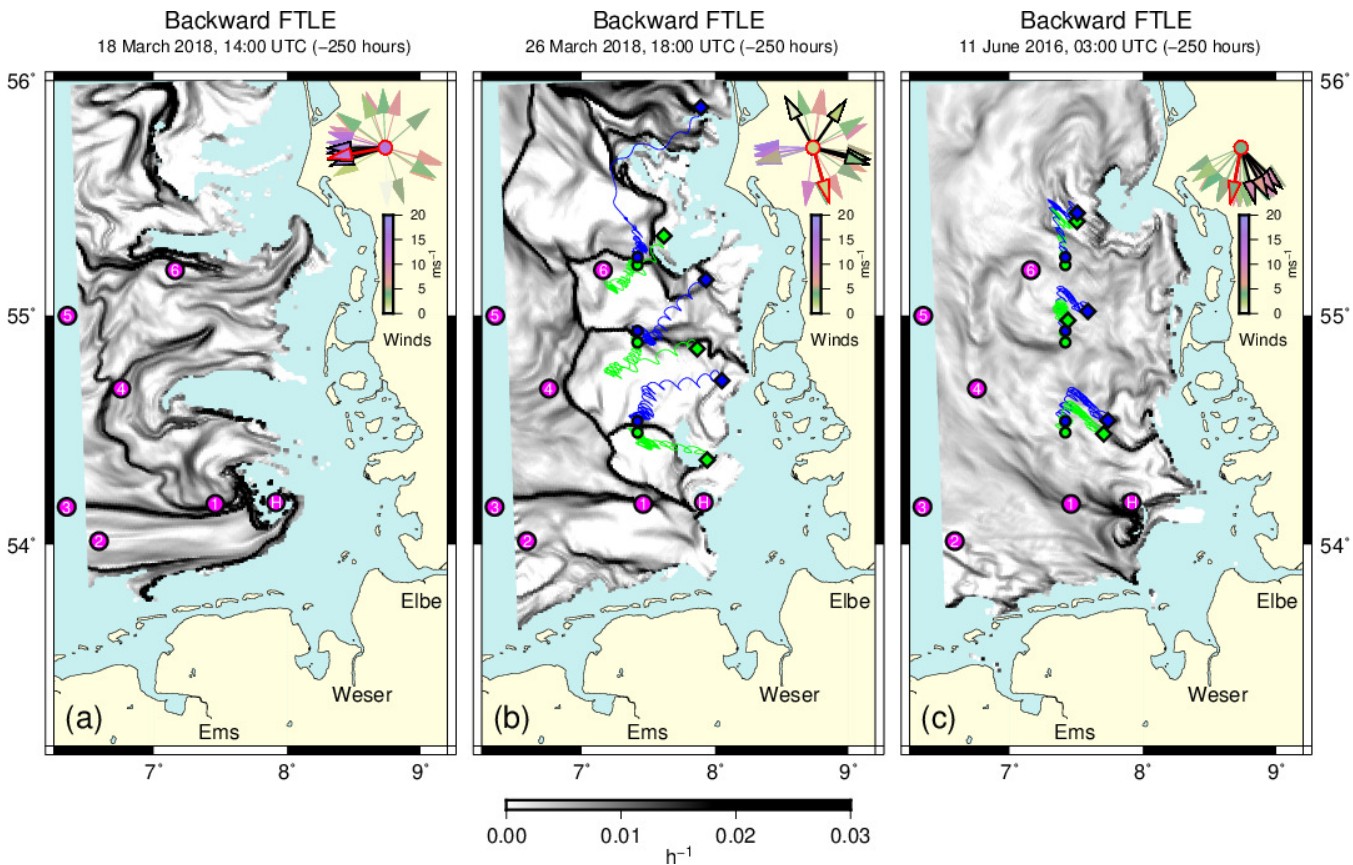

**Figure 3.** (a) Backward FTLE field (250 hours integration) for 18 March 2018 (14:00). Magenta circles indicate locations of MARNET stations (1-6) and of the island of Helgoland (H). (b) Backward FTLE field about eight days later (26 March 2018, 18:00). Three pairs of example trajectories (green/blue), started on either side of FTLE ridges, are shown. Trajectory release points are indicated by circles, end points by diamonds. (c) Example of a much smoother backward FTLE field on 11 June 2016 (03:00). For the purposes of comparison, backward trajectories were calculated from the same release points that were already used in panel (a). Vectors in the top right corner of each panel show past wind directions (modelled for the location of MARNET station 4) at ten hourly intervals. Different wind speeds are represented by different colours. Wind vectors at the times plots refer to are edged in red, those during the previous 50 hours are edged in black.

For comparison purposes, Fig. 3c shows the example of a much less structured FTLE field on 11 June 2016 (03:00) after persistently moderate winds from northerly directions. In this case sharp FTLE ridges are nearly absent. Overlaid to the FTLE field, Fig. 3c includes direct counterparts of the trajectories shown in Fig. 3b, released at exactly the same locations. Contrary to the situation in Fig. 3b, now all neighbouring trajectories are very much alike, mainly shifted in agreement with the slightly modified release points. A similar behaviour occurs also at the time of Fig. 3b when particles are released away from the FTLE ridges (see Fig. S1 in the supplement).

### 3.3 FTLE and measures of dispersion

29 February 2016 (Fig. 4a) provides another example of weak winds that follow more stormy conditions. At about 26 February, strong winds to the south of an atmospheric low make way for weaker winds under the influence of a high pressure system. Different from the previous example, however, the strong winds some days ago persistently blew from the west rather than from the east (see also the video provided in the supplement). Again a net of sharp FTLE ridges can be observed in Fig. 4a.

Figs. 4b and 4c analyse the situation in terms of statistical dispersion measures. Fig. 4b displays the spatial distribution of absolute dispersion. Each pixel in the plot represents the squared distance between the corresponding trajectory's release and end point. The plot reveals some sharp demarcations between zones with either broadly similar or at most smoothly changing drift velocities. A measure that directly concentrates on small scale changes in drift behaviour is two-particle relative dispersion (Fig. 4c). Its distribution closely resembles the FTLE field in Fig. 4a. Also the two maps of absolute and relative dispersion are in very good agreement, relative dispersion highlighting sharp transitions in the graph of absolute dispersion.

Fig. 4b includes some example trajectories. Test trajectories near the horizontal divide south of MARNET station 4 illustrate a stepwise change of advection speed, giving rise to the enhanced level of absolute dispersion for the hypothetical station located more to the south (green). Note that a pure change of drift direction, maintaining advection speed, would have affected relative but not absolute dispersion. Three additional trajectories (magenta), seeded at MARNET stations 1, 2 and 6, were included to just visualize spatial variability of transports.

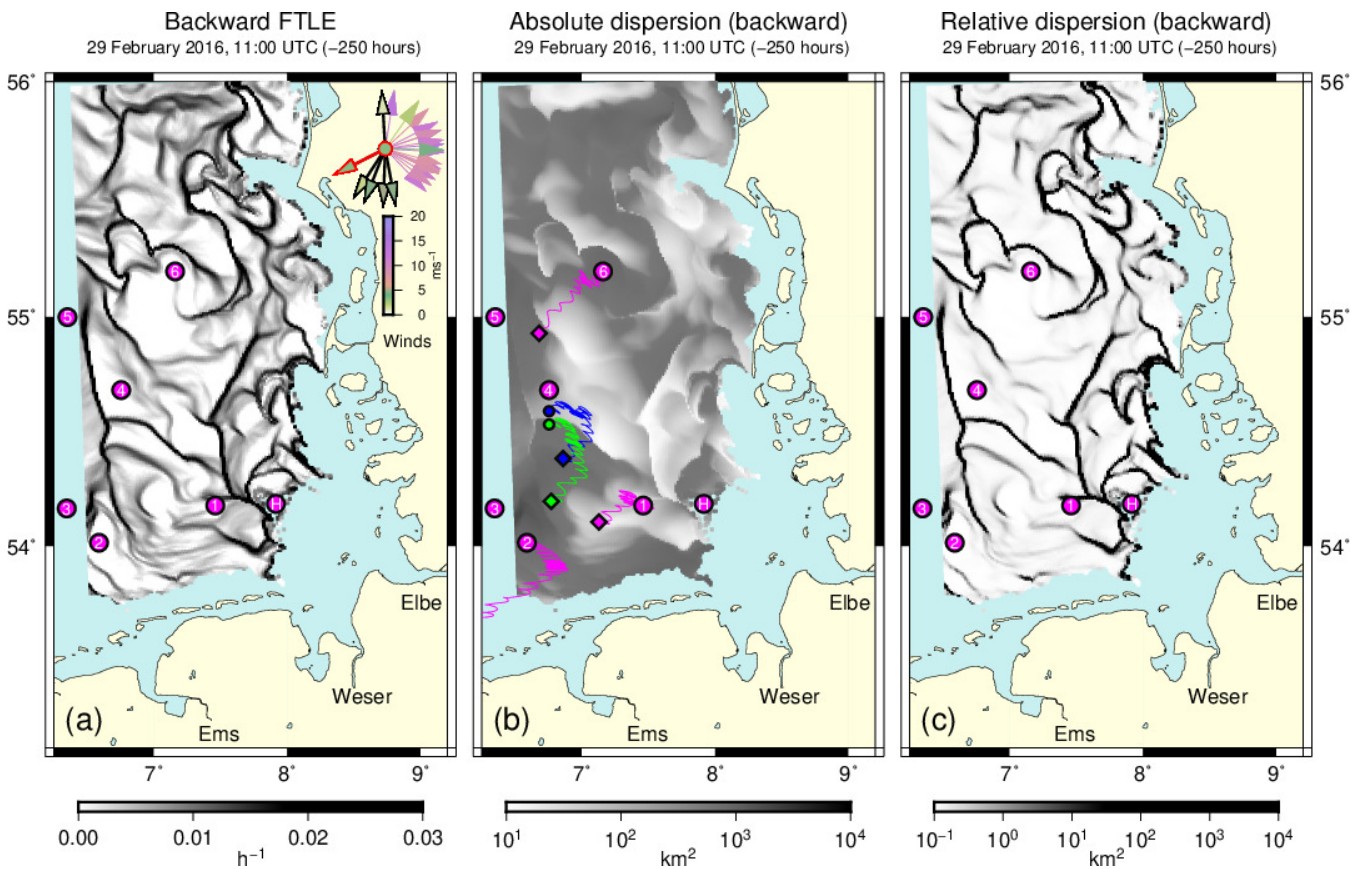

**Figure 4. (a)** Backward FTLE field (integration time 250 h) for 29 Feb 2016 (11:00). Magenta circles indicate locations of MARNET stations (1-6) and of the island of Helgoland (H). Vectors in the top right corner show past wind directions (modelled for the location of station 4) at ten hourly intervals. Different wind speeds are represented by a colour map. The wind vector at the time of the plot is edged in red, those during the last 50 hours are edged in black. **(b)** Absolute dispersion (see Eq. (12)) for the same time. Example trajectories shown were initialized at MARNET monitoring stations 1, 2 and 6 (magenta) and at two locations (small circles in blue and green) neighbouring MARNET station 4 to its south. Small diamonds indicate each trajectory's final location. **(c)** Distribution of relative dispersion (see Eq. (13)) for the same time.

### 3.4 Temporal development of FTLE fields

A video in the supplement, based on one FTLE field every 7 hours, shows the variability of FTLE ridges throughout the year 2016. The three panels of Fig. 5 are extracted from this video. They illustrate the development within the two week period 22 November to 6 December 2016.

Fig. 5a shows the situation after 10 days of mostly strong winds from between southeast and west. The FTLE field is much less compartmentalized than the fields in Figs. 3b and 4a, for instance. Instead it shows long FTLE ridges aligned in a meridional direction, resembling the pattern in Fig. 2a. On 23 November, winds blowing from southern directions change to winds from northern directions, which entails a reversal of the formerly pronounced cyclonic to an anticyclonic marine residual circulation[2]. The wind rose in Fig. 5b (27 November, 21:00) clearly separates a cluster of southern winds underlying Fig. 5a from the northern winds more recently. It can be seen how this transition of winds generates structures including also more east-west oriented ridges (Fig. 5b). After 3 December, a high pressure area with very low winds extends into the North Sea region. Under such calm conditions, the FTLE field shown in Fig. 5c suggests a development towards a more cellular structure.

---

[2]see https://www.bsh.de/DE/DATEN/Stroemungen/Zirkulationskalender/zirkulationskalender_node.html

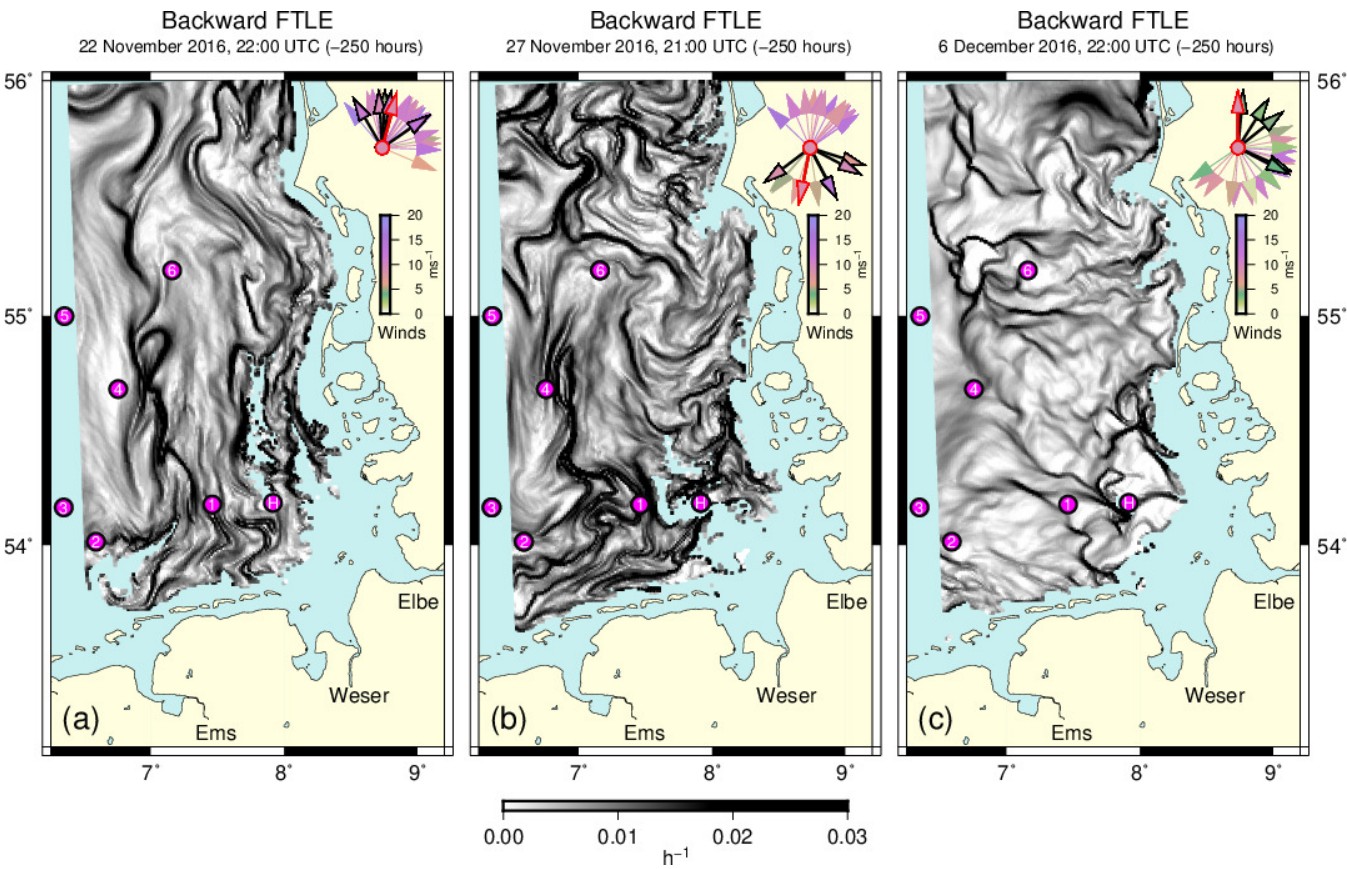

**Figure 5.** Three backward FTLE fields scheduled with approximately five and nine days in between, extracted from a video available in the supplement. Magenta circles indicate locations of MARNET stations (1-6) and of the island of Helgoland (H). Vectors in the top right corners show past wind directions (modelled for the location of station 4) at ten hourly intervals. Different wind speeds are represented by a colour map. Wind vectors at the times of the plots are edged in red, those during the last 50 hours are edged in black.

# 4 Discussion

Taking a monitoring perspective, this study focussed on the analysis of attracting LCSs, technically identified as repelling LCSs in backward simulations. Drift simulations based on BSHcmod surface layer currents revealed a pronounced time variability of LCSs, driven by changing wind conditions. Some LCSs, identified as ridges in the FTLE field, were found to be surprisingly sharp (see Figs. 3b or 4a), so that measurements at neighbouring locations might see water bodies with very different backgrounds. Similarly, at a given station, even a small wind-induced displacement of the FTLE field could substantially shift the origins of water bodies being probed. Being aware of such sensitivities can be relevant for a proper interpretation of observational data. Ridges in the simulated backward FTLE field convey the information in a clear and amenable way.

The general idea followed here differs from the objective of Ricker and Stanev (2020), for instance, who aimed at the identification of mean particle accumulation patterns (in forward mode) in the European northwest shelf on time scales of months or years. In the light of time-dependent, sometimes narrow FTLE ridges, a general characterization of monitoring stations in terms of their areas of influence seems difficult to achieve. Duran et al. (2018) derived climatological LCSs (cLCS) based on low-pass filtered velocity fields. These cLCSs could then successfully be applied for a description of quasi-steady transport patterns in the Gulf of Mexico. Marine currents in the German Bight area, however, are much more variable. Therefore, this study suggests the use of detailed numerical simulations to classify probed water bodies with regard to their presumable source regions. Similarly, detailed transport modelling could support the effective organization of field experiments. Not looking into the future, backward FTLE fields can be simulated already at the time when observations are actually taken. Such timely model based information on existing LCSs would allow for an adjustment of field campaigns to prevailing environmental conditions in the light of the data already gathered.

Examples shown suggest that in particular strong wind conditions trigger the occurrence of pronounced FTLE ridges. These ridges are often of considerable length and sometimes demarcate a net of closed subregions. They continue to exist for some time also under subsequent calmer wind conditions (e.g. Figs. 3a and 3b). Throughout this study, all FTLE values were calculated based on trajectories integrated 250 hours back in time. This integration time is much longer than few tidal cycles which, for instance, Orre et al. (2006) chose for analysing topographically constrained currents in a Norwegian fjord. Huhn et al. (2012) chose 24 hours for their study in the Ria de Vigo estuary, to prevent tracer particles from reaching the boundaries. In the present study, all trajectories that met water depths below 5 m were discarded. The long 250 h integration time implied that even when backward trajectories were started under calm atmospheric conditions, they could experience a storm event a couple of days ago. A very interesting observations is, however, that when integration time is reduced to just 50 h or even 25 h, the key FTLE ridges tend to become less sharp but do not disappear (see Fig. S2, which makes reference to the example shown in Fig. 3b). This finding is consistent with the fact that, according to Figs. 2a and 3b, neighbouring backward trajectories show high drift velocities towards the end of the integration period but at the same time a clear separation already right from the start. This indicates a certain memory effect after the storm has ceased.

In this study, FTLE fields were analysed on a 1 km grid, nearly matching resolution of the marine current fields. Computationally more demanding FTLE analyses on a finer grid would have enabled identification of structures even smaller than

the resolution of the underlying Eulerian hydrodynamic model. Such structures arise, however, from tracer simulations over longer distances (Huhn et al., 2012), thereby filtering more short-term features (Serra and Haller, 2016). As a result, a clear classification of kinematic LCSs in terms of mesoscale or submesoscale processes seems difficult.

In their study of Lagrangian transports in the Gulf of Mexico, Duran et al. (2018) found patterns of two-dimensional transports to arise from a mere confluence, i.e. normal attraction and tangential stretching without convergence. Lehahn et al. (2007) found satellite observations of chlorophyll filaments in the northeast Atlantic to well agree with simulated geostrophic transports, contracting at and stretching along material lines. Referring to Lapeyre and Klein (2006), they argue that an ageostrophic secondary circulation injecting nutrients from deeper layers may trigger further chlorophyll production. Similarly, Olascoaga et al. (2013, their Fig. 1) show a chlorophyll $a$ plume in the Gulf of Mexico to coincide with a divergence-free attracting LCS. At the smaller submesoscale, however, Hernández-Carrasco et al. (2018) found negative extremes of Lagrangian divergence to coincide with attracting LCSs identified as ridges in the backward FSLE field analysed from HF radar data. Also in the present study, the field of path-averaged divergence FDLD suggested the role of backward FTLE ridges as lines of convergence in coastal waters. This was explicitly shown for the example in Fig. 2, but pertains also to all other examples.

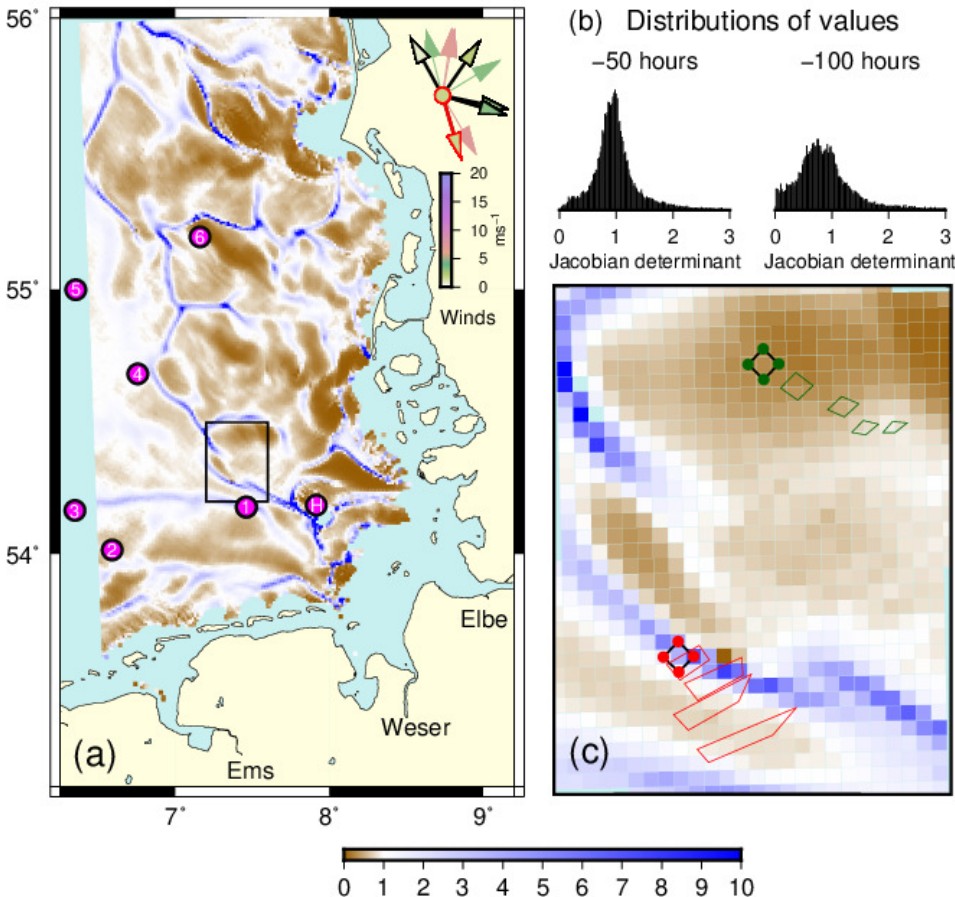

**Figure 6. (a)** Values of the Jacobian determinant (Eq. (3)) for 100 h backward simulations, initialized on 26 March 2018 (as in Fig. 3b). Few outliers exceeding the maximum value of the colour map were not specifically marked. Any negative values that occurred due to finite initial distances, were omitted in the plot. Wind directions during integration time are shown at ten hourly intervals in the top right corner of the panel. The actual wind vector is edged in red, those during the last 50 hours are edged in black. **(b)** Histograms of Jacobian determinant values for integration times -50 h (left) and -100 h (right), respectively. Note that the distributions' flat tales extend beyond the ranges of values being shown. **(c)** Zooming in on the subregion indicated by the black frame in panel (a), this panel shows the time evolution of two square patches, corners of which are made up by the initial locations (red and green dots) of the four trajectories needed to calculate the discretized deformation gradients (Eq. (2)) at the squares' centers. The developments of deformed quadrangles during the -100 h integration period are shown at 25 hourly intervals. Factors of area in- or decreases $\mu_1 \mu_2$ (Eq. (3)) are 1.56, 2.50, 3.40, 3.73 (red) and 0.96, 0.70, 0.42, 0.31 (green), where the last values equal the values shown in the map. Corresponding stretches $\mu_1 / \mu_2$ are 1.88, 3.01, 3.78, 5.52 (red) and 1.17, 1.52, 1.98, 2.36. (green).

Referring to the example addressed in Fig. 3b, Fig. 6a shows the corresponding field of the Jacobian determinant. A shorter integration period of only -100 h (rather than -250 h) was chosen to exclude the period of strong easterly winds that occurred about 8 days before the date of the analysis (compare the wind roses in Fig. 3b and Fig. 6a, respectively). Despite the shortened integration time, ridges in the backward Jacobian determinant field (Fig. 6a) well coincide with FTLE ridges (Fig. 3b). Values of the Jacobian determinant substantially deviate from the neutral value of one that corresponds with zero divergence (Fig. 6b). Values further spread with increasing integration time. To exemplify corresponding patterns of transport, Fig. 6c shows the development of two patches with an either increasing or decreasing area. At each time level, quadrangles are defined by the positions of the four trajectories that emerge from the locations used to calculate the discretized deformation radius (Eq. (2)) in their centre. In the more southern example, trajectories originate from locations on a ridge in the Jacobian determinant field. After 100 hours back in time, the initial area of the square has grown by a factor of 3.73, the value of the determinant shown in the map. By contrast, for the more northern example, the area decreases by a factor of 0.31. In both examples the drift behaviour is far from non-divergent.

In both cases, however, also substantial stretching occurs. Final stretches $\mu_1/\mu_2$ of the example squares in Fig. 6c amount to 5.52 (red) and 2.36 (green). To differentiate the effects of divergent flows from divergence-free repulsion, Huntley et al. (2015) introduced the decomposition of the FTLE measure into dilation rate $\Delta$ and stretch rate $\Sigma$ (see Eq. (8)). For the three examples presented in Fig. 2, Fig. 3b and Fig. 4, respectively, Table 1 provides the correlations between FTLE and either dilation rate $\Delta$ or stretch rate $\Sigma$. In each case, correlations are given for the three different integration times of -50 h, -100 h and -250 h. Correlations between FTLE and $\Delta$ or $\Sigma$ are both positive and of similar size, indicating that repulsion in the vicinity of backward FTLE ridges indeed occurs as a combination of divergence and stretching. In all three examples, it is interesting to note that the correlations between FTLE and dilation rate $\Delta$ increase with integration time, while correlations with stretch rate $\Sigma$ decrease. As a result, correlation of FTLE with dilation rate $\Delta$ is dominant for the -250 h integration time chosen in this study.

The temperature field in Fig. 2c reflects some effects of confluent and possibly also convergent currents. Surface temperature observations by remote sensing might possibly provide a means to confirm simulated FDLD fields. Schrum (1997) showed how the spatial extent of thermohaline stratified areas, a precondition for the occurrence of tidal mixing fronts, depends on wind forcing that possibly induces differential advection. In a recent paper, Chegini et al. (2020) provided a more detailed analysis of different processes that affect stratification and destratification in the German Bight area, including freshwater buoyancy input. Location of the Elbe River plume again depends on the wind driven residual circulation, which further substantiates the assumption of atmospheric forcing being the key driver for the generation, movement and extinction of German Bight LCSs.

Relatively persistent FTLE ridges related to the island of Helgoland, for instance, could possibly be relevant for sedimentation processes. However, movements of inertial tracers can substantially differ from those of fluid parcels, so that an analysis of ideal passive tracer trajectories is likely to be too simplistic in that context. The idea of LCSs has been generalized, however, to explicitly include the dynamics of inertial particles (Sapsis and Haller, 2009; Sudharsan et al., 2016; Günther and Theisel, 2017). This theoretical concept has successfully been applied on the scale of ocean eddies (Beron-Vera et al., 2015) but also on the very small scale of jellyfish feeding (Peng and Dabiri, 2009; Sapsis et al., 2011).

**Table 1.** Correlations between FTLE and its additive components $\Delta$ and $\Sigma$ (see Eq. (8)).

|  | $\tau$ | 12 Jun 2015 (Fig. 2) | 26 Mar 2018 (Fig.3b) | 29 Feb 2016 (Fig. 4) |
|---|---|---|---|---|
| corr (FTLE, $\Delta$) | -50 h | 0.41 | 0.59 | 0.68 |
|  | -100 h | 0.51 | 0.66 | 0.73 |
|  | -250 h | 0.65 | 0.72 | 0.77 |
| corr (FTLE, $\Sigma$) | -50 h | 0.69 | 0.63 | 0.57 |
|  | -100 h | 0.65 | 0.57 | 0.47 |
|  | -250 h | 0.52 | 0.50 | 0.37 |

For three examples presented in this paper, the table shows correlations obtained for backward
integration times $\tau$=-50 h, -100 h and -250 h, respectively.

Although numerical models can make observers aware of FTLE barries that move, disappear or newly arise under changing
environmental conditions, they can never provide a perfect surrogate nature. Guo et al. (2016) propose concepts to extend the
conventional analysis of deterministic FTLE fields and ridges to uncertain flow conditions. In a comparative study, Hufnagl
et al. (2017) found considerable discrepancies between the results from a large number of different North Sea tracer simulations
essentially based on vertical mean currents. Wiggins (2005) makes reservations that, as contrasted with many engineering
applications, the presence and interaction of very different scales in geophysical flows can restrict the possibility of simulating
detailed particle drift paths. Altogether, simulated FTLE field will always be imperfect. However, even in case of inaccurate
simulations, simulated FTLE fields will warn users about key sensitivities of specific model output. If an observation is taken
close to a simulated FTLE ridge, a simulated backward trajectory for this location must be used with due care.

This study did not address repelling LCSs in prediction (forward) mode. Drift simulations are important tools employed for
search and rescue (SAR), for instance (Breivik et al., 2013). Serra et al. (2020) proposed the use of objective Eulerian coherent
structures (OECS) in this context, a concept developed by Serra and Haller (2016) for being used when quick operational
decisions are to be made. According to Serra and Haller, OECSs can be understood as short-time limits of LCSs, applicable
for a time horizon of very few hours. More long-term forward simulations are feasible and can be afforded in the context of
ecosystem hindcasts, analysing larval transport and dispersal, for instance.

For surface drift simulations, additional uncertainties may arise from the necessity to specify the extent to which near surface
currents are exposed to a direct wind drag. Callies et al. (2017b, 2019) found that a successful simulation of observed drifter
trajectories needed BSHcmod surface currents to be augmented by a leeway of 0.6 % of 10 m winds. Besides a small direct wind
drag exerted on the drifters themselves, this leeway compensates for insufficient vertical resolution in the archived BSHcmod
surface current fields (representing a layer of 5 m depth). In addition, the leeway may also parameterize wave related Stokes
drift not being considered explicitly (Callies et al., 2017b; Sutherland et al., 2020; Staneva et al., 2021). Example forward FTLE

fields including a 0.6 % leeway are shown in Fig. S3. The examples show that FTLE ridges are modified but do not disappear when the smooth fields of a wind induced leeway are superimposed to marine currents. This conclusion directly translates to all the backward FTLE fields analysed in this paper.

## 5 Conclusions

The analysis of backward surface tracer simulations in the German Bight region revealed the intermittent presence of linear structures (LCSs) across which the past history of water bodies substantially changes. Such sensitive dependences of backward trajectories on tracer seeding positions, represented by narrow ridges in the FTLE field, could entail differences between in situ observations even at neighbouring locations. Therefore, an evaluation of spatially distributed in situ observations could benefit from the awareness of changing FTLE fields, analysed based on either numerical simulations or possibly high frequency (HF) radar observational data.

In the presence of narrow FTLE ridges, marked differences between observed and simulated tracer trajectories do not necessarily reflect poor model performance. If the location of a simulated LCS does not fully agree with reality, a tracer release point may come to lie on different sides of the separatrices in the model and in nature, respectively. In this case, a naive comparison of trajectories could much exaggerate inconsistencies. The same arguments pertain to a comparison of different drift models. Therefore, conventional model evaluations based on individual drift paths might be complemented with a comparison of simulated FTLE fields.

Examples illustrated the variability of German Bight surface layer LCSs under changing wind conditions. For a more comprehensive picture it could be useful to establish a method that allows to estimate the basic characteristics of backward FTLE fields from the recent history of atmospheric forcing. The examples studied suggest that model uncertainties occur particularly in the aftermath of storm conditions. Due to the presence of sometimes complex filamentary structures, a decomposition of FTLE fields in terms of a mean plus a number of weighted anomaly fields seems not very promising. A more qualitative classification of FTLE fields into a set of categories could be feasible. This issue is left to future research.

*Code and data availability.* The hydrodynamic data analysed in this paper were obtained from the repository of the Federal Maritime and Hydrographic Agency (BSH). For access to the archived results of the operational hydrodynamical model BSHcmod, please contact BSH (www.bsh.de). The Lagrangian drift code PELETS is available on request from the author.

*Video supplement.* A video is provided (FTLE_2016.avi) that demonstrates the temporal development of FTLE fields in the course of the year 2016, based on FTLE fields calculated every 7 hours.

*Author contributions.* The author performed all analyses and prepared the manuscript.

*Competing interests.* The author declares that he has no conflict of interest.

*Acknowledgements.* Drift simulations were based on BSHcmod currents provided by the Federal Maritime and Hydrographic Agency (BSH). Graphs were produced using the Generic Mapping Tools software (GMT) available from www.soest.hawaii.edu/gmt/. I would like to thank Rodrigo Duran for some helpful suggestions. I furthermore appreciate constructive comments by Jens Meyerjürgens and two anonymous referees.

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
