# Peer review of "Sensitive dependence of trajectories on tracer seeding positions - coherent structures in German Bight backward drift simulations"

_Ocean Science, 2020_

## Referee Comment (RC1) · Anonymous Referee #1 · 8 Oct 2020

The manuscript investigates Lagrangian metrics such as FTLE and absolute/relative dispersion using model results in the German Bight, showing the high variability of LCSs and enhanced space variability of trajectories close to ridges. The results are interesting but certainly not novel or unexpected, and the paper lacks in my opinion of clear focus and motivation. The author mentions several motivating applications, such as characterization or guidance for the observing system, but it is unclear how this would be carry out.

I think the paper needs an extensive revision or even better a re-submission, where the motivations and the elements of novelty are clearly indicated and developed. Also,

there are several specific points that need further clarification, as detailed in the following.

Main points

1) As mentioned above, there is an extensive literature showing the high sensitivity of particle trajectories to their seeding and the use of LCS to characterize it, so the results presented here are not new. I think the paper needs a novel angle, and a more specific motivation to make the present results new and interesting. I thought that the angle mentioned by the author in the Introduction regarding the characterization of an observing system composed of fixed points is interesting. But it needs more focus and more practical applications. For instance, could the results be used to quantify uncertainty at the stations using as proxies the distance from ridges? Or could they be used to indicate areas of influence of the stations, in terms of LCS patterns of dispersion properties? Investigating this type of questions would be very useful from the application point of view and could lead to new results.

2) Since the results are based on the BSH model outputs, it is very important that the model set up and its validation are adequately described, significantly improving Section 2.2. This is especially relevant since the model based results are envisioned to be used in support of the observing system, possibly also in real time. More specifically, has Lagrangian validation ever been performed using drifters? It is also important to be up front regarding model limitations. For instance, given the 1 km resolution we can expect that coastal submesoscale is only partially resolved at best. Also, if the model is hydrostatic, we cannot expect that near surface divergence processes are correctly described.

3) The description of the used techniques in Section 2.4- 2.5 should be improved, indicating also possible limitations and clarifying definitions. For instance, is the definition of FTLE in eq (3) valid in the case of 2-dimensional flows (as the text at line 120 seems to imply)? Also, what is the diference between eq (4) and (5) for dilation? From the

text (line 144-145), they seem to indicate the same thing, but it is unclear. Indeed the results in Fig.1b,c are quite different.

4) In general, I think that the text commenting the results should be more realistic throughout the paper. For instance the comparison between LCS ridges, and the 2 forms of dilation in Fig.1 (lines 188-195 and lines 312-13) is very positive, while I fail to see a good comparison between the figures. I do not see a "striking similarity" between Fig.1a and 1b, where the main North-South ridge is absent. The author acknowledges the clear difference between Fig.1b and 1c, but I do not understand the point of the comparison, given that the model itself is not well suited for this diagnostics. Also the comments on Fig.4 do not seem very grounded to me. In a case with very little gradients, except for the obvious coastal ones, as in Fig.4a and at some extent 4c, it is impossible to draw any meaningful conclusion.

5) Finally, and very importantly in my opinion, new diagnostics and metrics should be investigated, related to the observing system as mentioned in point 1). How can LCS be used to evaluate the observing system? How do LCSs vary on time? At which scales? Which proxies can we use to quantify these changes?

More specific points

The Introduction (and possibly also the title) should be re-written with more focus toward point 1) above. More in details, many phrases are unclear. Some examples are listed below - below line 20: "deficiencies of the underlying hydrodynamics...". Is this phrase indicating subgrid uncertainties or what? Deficiencies is certainly not the right word - around line 35. Discussion on local-versus nonlocal is not very precise. Indeed, local relative dispersion has been shown to be much faster at small scales and initial times than non local (Poje et al., 2014). It should also be clarified throughout the text wether the emphasis is on mesoscale or submesoscale dynamics

Section 2.1. It would be useful to mention from the beginning (lines 80-85) the geographical extension of the German Bight (lat/long are now mentioned at line 128 in

Section 2.4), and clarify that the area is depicted in all the figures.

Section 3, on results. The author shows 3 examples of LCS (Fig1, 2) for three different flow realizations and dates, 1 example of particle stats (Fig.3) for an other realization, and finally SST (Fig.4) for a mix of realizations. It would be better to focus on 3 cases only, and compare LCS with particle stats, as well as SST.

Section 4. provides a broad discussion on FTLEs and their applications, but there is no clear connection with the present results. Indeed, most of the information are more suitable for the introduction, and in any case should be trimmed and focused on the paper's goals

———————————————————

---

## Short Comment (SC1) · 13 Oct 2020

Dear Editor, dear Author, I appreciate the presented work on LCS in the German Bight, North Sea very much. Its application and usage for oceanographic purposes is not new but has not been applied to this region, yet. However, several major concerns arose during the reading of the manuscript. I find the presented topic worth it to be published and thus I want share these concerns which will hopefully be considered in case of a manuscript revision. The order does not represent their priority:

- What is the reason for the choice of the time periods used for LCS analyses considering the high variability of the LCS structures? Are they representative oceanographic

states of the German Bight? Why not using temporal averaged fields? - The citation of earlier studies of the German Bight with focus on Lagrangian methods would be very useful in the introduction. This is essential also with respect to a comparison of LCS and the results of earlier studies (see also below). For example, Ricker and Stanev (2020, doi:10.5194/os-16-637-2020) show a particle accumulation pattern in the German Bight being similar to the major LCS pattern with north-south orientation. - With respect to other regions, some important works (e.g., by G. Haller) have been mentioned. Nevertheless, I'm missing representative studies with a more practical and oceanographic background; an example is I. Rypina. Such studies would probably enable direct comparisons of the results between different regions of interest. - Why is not the density shown instead of T and S? Why are only instantaneous fields shown? The LCS are obtained from a 250-h period. In addition, a validation of the surface temperature fields would be nice. Some of them seem to be unrealistic. - Page 2, line 50: Examples of the German Bight should also be cited, e.g. Meyerjürgens et al. (2020). - I'm missing an explanation of the underlying dynamics favouring the LCS structures. There is no discussion taking into account the physical oceanography of the German Bight. What about frontal dynamics? - The hydrodynamic model: The number of vertical layers is missing. How can the use of a 5 m surface layer be reasoned if the region of interest has depth range of 20-40 m? Is the setup able to realistically resolve vertical processes which, probably, lead to pronounced LCS patterns? - Why are the three quantities (FTLE, FDLD and dilation rate) chosen if they are so similar? Is there a physical reason? What is expected to be different among them? - Are there more practical conclusions that can be drawn from the study? What about oil and floating marine litter? I'm missing the actual aim of the study which should be stronger highlighted in the introduction. - Page 3, line 65: Stanev et al. (2019, doi:10.1016/j.csr.2019.03.003) is a more recent example demonstrating the complete reversal of the North Sea currents due to atmospheric forcing. - Page 3, line 66: However, distinct long term current patterns do exist in the German Bight. They are important for long term tracer dynamics. There are several examples in the literature also in case of the North Sea. - Some figure descriptions are in the text and not in the figure captions - Is there a reason for the 1 km seeding grid? - What is the link between observations and modelling? Which role do the MARNET stations play? Is there a proof of different water masses separated by LCS in the station data? - It is mentioned that tides have a strong influence on the LCS and dispersion patterns. What does this influence mean for the analysis of FTLE and dispersion? - I'm missing a clear structure in the discussion. Technical details are also included. As mentioned before, the link between LCS and the hydrography should be discussed in detail. In addition, the discussion also lacks of Lagrangian studies (GPS drifters as well as simulations) of the German Bight and how the presented results could be related to them.

Yours sincerely, Jens Meyerjürgens

---

## Author Comment (AC2) · 15 Oct 2020

Dear Jens Meyerjürgens,

I appreciate your comments on this study. Please find below my answers to the main points you raised.

**Why not using temporal averaged fields?**
Due to the strong variability of spatial structures, average fields are very smooth, the following distribution gives an example (annual mean for 2016):

[Figure]

A LCS at a given time depends on a specific recent history of wind forcing. Winds are not constant over longer times, therefore I am not sure that FTLE fields specified for different (constant) wind directions would be very informative. But it might nevertheless be an interesting experiment.

**The citation of earlier studies of the German Bight with focus on Lagrangian methods:**
You are right, the recent paper by Ricker et Stanev (2020) is a useful reference because the authors also discuss the relevance of wind forcing.

**I'm missing representative studies with a more practical and oceanographic background; an example is I. Rypina. Such studies would probably enable direct comparisons of the results between different regions of interest.**
Do you mean articles like Serra et al. (2020), *Search and rescue at sea aided by hidden flow structures*, co-authored by I. Rypina? That could indeed be an interesting. A comparison of different regions, however, would definitely be beyond the scope of this paper.

**Why is not the density shown instead of T and S?**
This study was based on archived data, the hydrodynamic model wasn't re-run to generate additional information.

**I'm missing an explanation of the underlying dynamics favouring the LCS structures. There is no discussion taking into account the physical oceanography of the German Bight. What about frontal dynamics?**
For analysing frontal dynamics, for instance, one should go back to the original model rather than just archived model output. This study suggests using data that are available operationally and could be used even for real time analyses. FTLE fields were produced for a couple of years with one field every 7 hours. From this data base, then few example fields were discussed in the paper. A subsequent dynamical study might focus on an analysis of different LCSs with possibly different dynamic backgrounds. Also using different hydrodynamic models would be interesting.

**The hydrodynamic model: The number of vertical layers is missing. How can the use of a 5 m surface layer be reasoned if the region of interest has depth range of 20-40 m? Is the setup able to realistically resolve vertical processes which, probably, lead to pronounced LCS patterns?**
In the German Bight area the model has a maximum of 24 vertical layers. Only for archiving these data were interpolated to another grid in which the top layer now combines results from different finer layers. The model dynamics, however, were implemented on a grid sufficiently fine to resolve all relevant processes.

**Why are the three quantities (FTLE, FDLD and dilation rate) chosen if they are so similar?**
FTLE and dilation rate are indeed very similar and dilation rate could possibly discarded. However, FTLE and FDLD are calculated quite differently and it is nice to see that results are so similar.

**Are there more practical conclusions that can be drawn from the study? What about oil and floating marine litter?**
It is hard to draw general conclusions when spatial FTLE patterns are so variable. The conclusion with regard to monitoring is that interpretation of actual measurements from multiple stations should take into account any substantial LCS that possibly separates one in situ station from another close by. I am not sure whether such considerations could affect the analysis of marine litter distributions, where high resolution monitoring data are missing.

**Page 3, line 66: However, distinct long term current patterns do exist in the German Bight. They are important for long term tracer dynamics. There are several examples in the literature also in case of the North Sea.**
As you say, this applies to long term behaviour. On a short term scale, however, currents and residual currents are very variable.

**Is there a reason for the1 km seeding grid?**
The resolution seems reasonable in the light of the spatial resolution of the underlying hydrodynamic model.

**What is the link between observations and modelling? Which role do the MARNET stations play?**
Here, the MARNET stations were shown to facilitate the description of LCS locations and a comparison of different situations. From an applied perspective one may be interested to see which stations have probably higher chances to be close to certain LCS structures. For this purpose, however, one has to look into longer times series of FTLE distributions (see the video in the supplement, for instance).

**Is there a proof of different water masses separated by LCS in the station data?**
No, the present analysis refers just to surface currents.

**It is mentioned that tides have a strong influence on the LCS and dispersion patterns. What does this influence mean for the analysis of FTLE and dispersion?**
Tidal movements are obviously important in combination with stirring effects of topographic structures, for instance near the island of Helgoland.

**The discussion also lacks of Lagrangian studies (GPS drifters as well as simulations) of the German Bight and how the presented results could be related to them.**
I am afraid that so far the data available in the German Bight area do not enable the discussion you are asking for. Callies et al. (2017b, 2019) successfully employed BSHcmod surface currents (also used here) to simulate observations from drifter experiments. These

studies are mentioned in the manuscript. However, the very limited number of drifters is definitely not sufficient to identify LCSs. In any case, it can never be expected that model results provide a perfect surrogate reality. Crosschecking LCSs obtained from different models and probably also from radar observations would be very useful.

Kind regards
Ulrich Callies

---

## Referee Comment (RC2) · Anonymous Referee #2 · 21 Oct 2020

Here, I try to provide my comments about this manuscript.

In general, the topic is appropriate for this journal, and I'm not aware of a similar study for the German Bight, which it makes it worth of publishing.

The used tools (PELETS in combination with surface current velocities obtained from BSH) are state-of-the-art and appropriate. Furthermore, the different quantitative measures to identify Lagrangian Coherent Structures (LCS) or objects with similar meaning are appropriate.

I must admit I'm not an expert for LCS and similar structures. However, reading the article and the theory included was very informative and helpful for me. Furthermore,

using more the one quantitative measure is quite illustrative.

Major comments and questions:

- Tides are very important for this region. Might it be possible to give some idea how and if the FTLE structures will change when tides are not subtracted from the currents. So, what might happen when not using residual currents.

- It was very illustrative to sea how variable the FTLE structures are in time. What would happen, if one considers time averaged FTLE-fields. Perhaps averaged over a season or half a year. Will there be stationary or more robust FTLE ridges visible?

- Instead of showing the temperature and salinity fields, perhaps one could use the density distribution. Would this do any difference to the interpretation of the relationship between FTLE ridges and T/S fields.

Some minor comments and questions:

- How do you calculate the residual currents from the BSH data? The current velocities of the BSH include tides, which are important for this region. Are the results dependend on the way these residual currents are calculated?

- Do you think it might be helpful to plot the absolute value of the gradient of temperature or salinity instead of the pure fields. So, one could see the sharp fronts more clearly.

Some direct comments to the text:

- Line 13: Gap at the end of sentence; (2017). Stanev . . .

- Line 143: The year of the Huntley citation is missing

- Equation (5): It is divergence of the 2d surface velocity field, isn't it?

- Figure 2a: The endpoint of the red drifter starting at 54 30' is missing?

- Perhaps one could also include the river Ems to the plots.
- Figure 4, caption: . . . times of the FTLE fields . . . ?

- Additional figure S3: The unit of salinity should be different from PSU. Perhaps no unit or g/kg

---

## Short Comment (SC2) · 17 Dec 2020

**Rodrigo Duran**

rodrigo@oceanresearch.xyz

Received and published: 17 December 2020

Reading the discussion of this paper, I found a fundamental error that has apparently gone unnoticed so far. In lines 312–320, the author argues that:

"In this study, analysed structures were remarkably consistent for fields of FTLE, FDLD, dilation rate or measures of dispersion. Differences between the FTLE and FDLD fields discussed by Huntley et al. (2015) could not be seen on the spatial scale considered. Fields of path-averaged finite-time Lagrangian divergence FDLD corroborate the role of backward FTLE ridges as lines of convergence (see Fig. 1). This relationship agrees with the results of many oceanographic studies. Olascoaga et al. (2013, their

Fig. 1), for instance, provide an example of how a chlorophyll a plume in the Gulf of Mexico coincides with an attracting LCS. Lehahn et al. (2007) found satellite observations of chlorophyll filaments in the northeast Atlantic to well agree even with simulated geostrophic transports, contracting at and stretching along material lines. Referring to Lapeyre and Klein (2006), Lehahn et al. argue that an ageostrophic secondary circulation injecting nutrients from deeper layers may trigger further chlorophyll production."

The author implies that FTLE detects confluence caused by convergence (along-path sampling of the negative divergence of a horizontal velocity). In reality, the attraction identified by FTLE is often due to confluence with negligible divergence, or even confluence in the presence of positive divergence. Indeed, the author himself mentions examples of divergence-free velocity fields. The Olascoaga et al. paper uses altimeter velocity, and the geostrophic velocity in Lehahn et al. will similarly have negligible divergence. These examples are therefore counterexamples to the author's conclusion. An example of confluence in the presence of positive Eulerian divergence can be found here (Fig 4): https://www.nature.com/articles/s41467-020-16281-x

If the author wishes to suggest that geostrophic (or more generally divergence-free) strain may induce ageostrophic circulation, then that is fine, and it has a precedent, e.g. https://www.nature.com/articles/s41467-019-10883-w.

However, to diagnose whether the attraction detected through FTLE is due to divergence-free confluence or convergence, additional computations are required. One can quantify the along-path change in area, as described in appendix A of this supporting information: https://static-content.springer.com/esm/art%3A10.1038%2Fs41598-018-23121-y/MediaObjects/41598\_2018\_23121\_MOESM1\_ESM.pdf (The article can be found at https://www.nature.com/articles/s41598-018-23121-y note this is an additional example of divergence-free confluence.)

By computing the Jacobian determinant (fractional change of area) of the flow maps used to compute LCS, we can diagnose when along-path convergence is negligible, in
which case attraction is due to divergence-free confluence. However, if the change-ofarea is substantial, the relative contributions of convergence and confluence still need to be quantified. A Helmholtz decomposition of the velocity may be helpful, yet going this route poses additional challenges.

Relating confluence with convergence without proper diagnosis is not uncommon, and therefore requires clarification as we work to understand the influence of submesoscales on dispersion. Please consider accommodating my comments in your paper.

Disclaimer: I did not read the full paper.

---

## Author Comment (AC5) · 18 Dec 2020

Dear Rodrigo Duran,

thanks your comments on this manuscript. I fully agree with your statements that "attraction does not imply convergence" and also that additional computations are required "to diagnose whether the attraction detected through FTLE is due to divergence-free confluence or convergence". Indeed these additional computational computations were already conducted in this study, computing the Finite-Domain Lagrangian Divergence (FDLD) (see section 2.5). Corresponding results are shown in Fig. 1b. You probably missed that when reading just the discussion (see your disclaimer). Which

means that the issue should be addressed more clearly in the discussion section.

In conclusion, the error you mention does not really exist. However, in the revised manuscript, the fact that generally attraction does not imply convergence will be stated more clearly. It will be emphasized that convergence along FTLE ridges is an extra outcome of this study rather than something that must necessarily occur.

Best regards

Ulrich Callies

---

## Short Comment (SC3) · 21 Dec 2020

Ulrich,

Thanks for considering my comment and planning to revise the paper to avoid suggesting that attraction implies convergence in general. Perhaps you can consider exchanging the use of "convergence" for "attraction" throughout the manuscript. For example, in the abstract where you mention: " A corresponding spread of backward simulations implies convergence in the forward passage of time." In this sentence, "attraction" instead of "convergence" would be correct.

[Figure]

Regarding attraction implying convergence in your study in particular, I would like to clarify a few points:

1) I did notice the FDLD computations, apparently only compared to one set of FTLE, while the paper describes other FTLE. Perhaps you can clarify if FDLD always coincides with FTLE in your study.

2) Note that an advantage of the method I suggested in my previous comment is that the sampling of along-path divergence happens along the same trajectories used to compute LCS. You mention you use the same "pixels" (caption of Fig. 1), it is unclear if you are using the same trajectories. In backward time and near attracting FTLE, small differences in initial positions may cause large differences in trajectories, as the title of your paper suggests. It, therefore, becomes unclear how the two sets of trajectories compare, although I would not be surprised if the results hold either way.

3) By quantifying along-path divergence one can arrive at one of two conclusions:

a) Divergence is negligible

b) Divergence is not negligible.

Concluding that FTLE is due to convergence just because option b) is true would overlook the possibility that attraction, at least in part, originates from confluence.

Here are a few thoughts along those lines:

In geophysical flows, it is possible, perhaps even likely, that when b) holds, then attracting LCS are due to the effect of both confluence and convergence—especially at time and space scales above a few days and the deformation radius, respectively. Consequently, there is a need to quantify individual contributions. Note that in figure 1, the FTLE and FDLD do have a remarkable resemblance, but there are also some locations where intense FTLE coincides with negligible or positive FDLD.

What does it mean that FTLE magnitude is often about two or three times larger than

FDLD (both are 1/hour)? (This is often the case when FTLE ridges and negative FDLD coincide).

Is it possible that the strain from confluence induces the ageostrophic circulation that FDLD detects? Note that the FDLD may be a localized consequence of confluence, e.g., see the schematic in figure 3 of https://www.nature.com/articles/s41467-019-10883-w Are FDLD ridges collocated to FTLE? How well do FTLE and FDLD ridges intersect in space and time?

These are all just possibilities that, in my view, need consideration before concluding with certainty that in your study, attraction implies convergence. I hope to have demonstrated that the correlation between FDLD and FTLE does not necessarily implicate FTLE causation by FDLD.

Thanks again, and best regards, Rodrigo.
* * *

---

## Author Comment (AC6) · 14 Jan 2021

Dear Rodrigo,

thanks for your constructive comments.

Using "attraction" instead of "convergence" actually seems to be a good option to avoid confusion.

The relationship between FTLE and FDLD turns out to be very stable, it occurrs for all examples. You are right, that must be stated more clearly.

And yes, the along-path divergences were calculated based on the same trajectories that were also used for LCS specification.

The revised manuscript will better address the relationship between confluence and convergence. Stretching plays an important role indeed, although its clear quantitative evaluation against convergence seems difficult. An evaluation could compare dilation and stretch, following the concept of Huntley et al. (2015). Nevertheless, the proper interpretation of such results based on the Jacobian's singular values is not always simple.

The figure below shows Jacobian determinant values for the example of 26 March 2018 (Fig. 2a in the original manuscript). Values of the Jacobian were specified for a 100 h backward integration period. The right panel zooms in on the black frame in the left panel. For two example locations it illustrates the time evolution of the four surrounding trajectories needed to calculate the discretized Jacobian at the median point of their initial locations (red or green dots). Contours of the emerging deformed quadrangles are shown every 25 hours. In agreement with the value of the determinant, the area of the initial square increases in the one case (final value of the determinant is 3.7) and decreases in the other (final value: 0.31). Stretching, however, defined as the ratio of the larger and the smaller singular value is substantial in both cases (5.5 and 2.4, respectively). As this ratio can never be smaller than one, some stretching will always be analyzed.

[Figure]

Nevertheless, you are right that attraction in part originates from confluence. The fact that both confluence and convergence contribute to the FTLE field could be substantiated considering the correlation between either dilation or stretch rate and the FTLE. Here, both correlations were found to be comparable in size. An interesting aspect is, however, that while correlation between FTLE and dilation rate tends to

increase with longer integration time, correlations between FTLE and stretch rate seem to decrease. For the integration period of 250 hours, used in the paper, linear correlation of FTLE with dilation rate tends to be larger than correlation with stretch rate. This will be substantiated for examples already discussed.

To strictly exclude any potential problems with the limited vertical resolution of the archived hydrodynamic currents, I decided to skip all trajectories that encounter a water depth below 5 m (instead of 0 m at the coast line) at any time during their journey. As a result, the study now clearly focusses on open sea conditions.

Could the FDLD be a localized consequence of confluence? This might be the case, but an in depth analysis of this 3D effect would need going back to the original hydrodynamic model instead of using coarsened model output available from the BSH archive. It is also to be emphasized that bathymetric effects are important for the effects observed in the German Bight coastal area.

A general evaluation of how well FTLE and FDLD ridges do intersect in space and time seems difficult due to the weather dependent occurrence of very different situations (see the video in the supplement) with ridges being either smooth or sharp. Examples show, however, that correlation between FTLE and FDLD (or dilation rate) seem to increase for those examples with particularly sharp FTLE ridges.

Thanks again for your very helpful comments!

Best regards,
Ulrich

---

## Author Comment (AC1)

Dear Referee,

to start with, let me thank you for your review and the time you spent on the evaluation of the manuscript. Below you will find my preliminary answers to some major points you raised.

Of course you are absolutely right that "there is an extensive literature showing the high sensitivity of particle trajectories to their seeding and the use of LCS to characterize it". However, the results of these studies are always site specific and I am not aware of such results for the German Bight area. The present study does not introduce the identification of Lagrangian coherent structures (LCS) as a new technical approach but rather describes LCSs that might be relevant for the proper interpretation of German Bight monitoring data. I do not agree that the shapes of the large LCSs, certainly also constrained by the bathymetry in this specific region, are trivial or were foreseeable.

Unfortunately, in its present version the manuscript does not explicitly state that the underling 3D model BSHcmod is baroclinic. The specific model was chosen for two main reasons:

- **First**, the model has been run operationally by the German authority BSH for many years, practical experiences were gained also in the context of search and rescue. Your question *"has Lagrangian validation ever been performed using drifters?"* can be answered in the affirmative. Two studies on this issue (Callies et al. 2017b, 2019) are referred to in the manuscript. However, it might probably be good to mention these studies also in the context of the model description.
- Second, the operational model was chosen exactly for the reason you mention in our review "... since the model based results are envisioned to be used in support of the observing system, possibly also in real time." BSHcmod results are reliably available on an everyday basis. Of course, higher spatial resolution would be useful, but such simulations are not available operationally.

I am puzzled as to why you say that "*I do not see a "striking similarity" between Fig.1a and 1b, where the main North-South ridge is absent.*" Actually, in Fig. 1b the FTLE ridge from Fig 1a is very clearly reproduced in terms of a line (green) of convergence. This and other lines of convergence are more or less perfect copies of the FTLE ridges (black) in Fig. 1a. The agreement is indeed striking.

I agree, however, that the display of temperatures in Fig. 4 is probably problematic. LCS related temperature differences exist, but of course they are small relative to gradients that occur towards the coast. Probably a more specialized colour scale should be chosen to resolve and highlight the details of specific interest. Nevertheless, I think that even now the structures in the FTLE fields can be identified also in the temperature fields.

You would like to see answers to the following questions: "How do LCSs vary on time? At which scales? Which proxies can we use to quantify these changes?" The first question can hardly be answered based on a necessarily limited number of figures in the manuscript. This is why a video was provided in the supplementary material, showing the evolution of FTLE fields over one full year. This video also answers your second question. However, probably an extra paragraph should be added to better summarize features seen in the video. Your third question is the most difficult one. It is addressed in the last paragraph of the conclusions. It would be extremely useful if LCS could be predicted via a simple dependence on atmospheric forcing. However, as the LCSs may depend on wind histories over extended periods of time, such a relationship cannot easily be established. This is why the manuscript leaves this aspect for future research.

You are asking for a more practical application of the results. I fully agree with your statement that "... the angle mentioned by the author in the Introduction regarding the characterization of an observing system composed of fixed points is interesting." Contributing to better interpretation of monitoring data is indeed the main idea of this study and the text should probably lay even more

emphasis on this application. However, a comprehensive example data analysis would have to involve a proper discussion of many different aspects, including all types of observational errors and unresolved turbulent processes on a subgrid scale, for instance. This would definitely go beyond the scope of this study. The lack of such specific application is also the reason why monitoring was not mentioned in the title of the manuscript.

I hope my remarks can clarify some points that possibly were not described clearly enough.

**Ulrich Callies**

---

## Author Comment (AC3)

**I would like to thank the referee for the time spent on the thorough review of this paper. In the following, referee comments are given in blue and responses in black.**

The manuscript investigates Lagrangian metrics such as FTLE and absolute/relative dispersion using model results in the German Bight, showing the high variability of LCSs and enhanced space variability of trajectories close to ridges. The results are interesting but certainly not novel or unexpected, and the paper lacks in my opinion of clear focus and motivation.

I am not aware of any analysis of LCSs in the German Bight area. Results of such studies will always be site specific and I would not agree that the shapes of the LCSs obtained, partly constrained by the bathymetry in this specific region, are trivial or were foreseeable.

**The author mentions several motivating applications, such as characterization or guidance for the observing system, but it is unclear how this would be carry out.**

'Guidance for the observing system' suggests that the distribution of LCSs would be used for a characterization of certain locations. I revised the discussion to make clear that this is not the main objective of the study. The idea is rather that detailed simulations addressing a specific period of observations might support an analysis of these data, helping to better distinguish between real changes in a system and a simple shift in the range of influence. This point is now better addressed in the discussion.

I think the paper needs an extensive revision or even better a re-submission, where the motivations and the elements of novelty are clearly indicated and developed. Also, there are several specific points that need further clarification, as detailed in the following.

**Main points**

**1) As mentioned above, there is an extensive literature showing the high sensitivity of particle trajectories to their seeding and the use of LCS to characterize it, so the results presented here are not new.**

You are right, a large body of literature on LCSs already exists. However, results of practical studies are always site specific and I am not aware of any such FTLE study for the German Bight area. The present study describes LCSs that possibly affect the meaning of German Bight monitoring data. I wouldn't say that the shapes of these large LCSs, also constrained by the bathymetry in this specific region, are trivial or were foreseeable.

I think the paper needs a novel angle, and a more specific motivation to make the present results new and interesting. I thought that the angle mentioned by the author in the Introduction regarding the characterization of an observing system composed of fixed points is interesting. But it needs more focus and more practical applications.

With regard to the characterization of an observation system, please see my previous response. In this study, the focus is on hydrodynamic simulations being used in support of data analysis. I hope that this point is now better explained in both introduction and discussion. Optimizing the sites of an observation network would be another interesting task, but possibly more challenging. The analysis of a specific data set would go far beyond the scope of the present paper, needing a discussion of many specific problems related to the data of interest.

For instance, could the results be used to quantify uncertainty at the stations using as proxies the distance from ridges? Or could they be used to indicate areas of influence of the stations, in terms of LCS patterns of dispersion properties? Investigating this type of questions would be very useful from the application point of view and could lead to new results.

I see your point. However, in the light of high variability of the LCSs it seems difficult (if not impossible) to define such characteristic areas of influence for specific stations. Areas observations are representative for much depend on the recent history of winds. Therefore, in the revised manuscript now all figures provide a summary of wind conditions over the last 10 days . The revised discussion now refers to this additional information. Although pronounced LCSs are initiated by strong winds, they may survive and even sharpen under subsequent calm wind conditions. Instead of a static characterization of monitoring stations, a model based representation of wind driven, variable hydrodynamic conditions could be useful to support interpretation of monitoring data.

2) Since the results are based on the BSH model outputs, it is very important that the model set up and its validation are adequately described, significantly improving Section 2.2. This is especially relevant since the model based results are envisioned to be used in support of the observing system, possibly also in real time.

The description in Section 2.2 has been extended, it now provides more detailed information about the model.

BSHcmod has been in operational use for about two decades, with revisions being applied where needed and possible. Practical experiences were gained also in the context of search and rescue. The operational model was chosen exactly for the reason you mention: BSHcmod results are reliably available on an everyday basis. Of course, higher spatial resolution would be useful, but for the moment such more expensive simulations are not available operationally.

**More specifically, has Lagrangian validation ever been performed using drifters?**

Yes, an identical model setup was successfully used for an evaluation of drifter experiments. References to two recent studies (Callies et al. 2017b, 2019) have been added at the end of Section 2.2.

**It is also important to be up front regarding model limitations. For instance, given the 1 km resolution we can expect that coastal submesoscale is only partially resolved at best.**

Thanks, this is indeed an important aspect. It is now addressed in the discussion (starting at line 304). A clear classification with regard to spatial scales seems hardly possible, as even large scale currents may generate LCS on a scale smaller than resolution of the grid used for underlying hydrodynamic simulations. This would need, however, the FTLE field being defined with higher resolution, which was not been explored in the present study.

**Also, if the model is hydrostatic, we cannot expect that near surface divergence processes are correctly described.**

The model is hydrostatic, but in the absence of significant bathymetric variations implying high vertical velocities, this shouldn't be a major restriction. The present analysis focusses on open sea conditions, where currents should be reliable (see former studies by Callies et al. (2017b) and Callies et al. (2019)).

**3) The description of the used techniques in Section 2.4- 2.5 should be improved, indicating also possible limitations and clarifying definitions. For instance, is the definition of FTLE in eq (3) valid in the case of 2-dimensional flows (as the text at line 120 seems to imply)?**

Section 2.4: It is not fully clear to me which kind of limitations and clarifying definitions this comment refers to. I think that all equations needed are already there. The formalism deals with a kinematic description of flow motions (according to Haller (2015): "...mimicking experimental flow visualization by tracers") so that there are no limitations with regard to the underlying physical dynamics. The

whole study deals with two-dimensional surface currents, which is now stated more explicitly: "Considering two-dimensional surface currents, this tensor is also two-dimensional."

Section 2.5: Contains Eqs. (4) and (5), please see my response to your next point.

Also, what is the difference between eq (4) and (5) for dilation? From the text (line 144-145), they seem to indicate the same thing, but it is unclear. Indeed the results in Fig.1b,c are quite different.

The dilation rate, defined in the former Eq. (4) has been removed from the paper in order to not overload the presentation.

4) In general, I think that the text commenting the results should be more realistic throughout the paper. For instance the comparison between LCS ridges, and the 2 forms of dilation in Fig.1 (lines 188-195 and lines 312-13) is very positive, while I fail to see a good comparison between the figures. I do not see a "striking similarity" between Fig.1a and 1b, where the main North-South ridge is absent.

I do not understand why you do not see the North-South ridge in Fig. 1b. Actually, the FTLE ridge from Fig. 1a is very clearly reproduced (a more or less perfect copy) in terms of a (green) line of convergence. The good agreement is really striking.

**The author acknowledges the clear difference between Fig.1b and 1c, but I do not understand the point of the comparison, given that the model itself is not well suited for this diagnostics.**

Fig. 1c showing the dilation rate has been removed and is now replaced by the temperature distribution (formerly Fig. 4a, see above).

Also the comments on Fig.4 do not seem very grounded to me. In a case with very little gradients, except for the obvious coastal ones, as in Fig.4a and at some extent 4c, it is impossible to draw any meaningful conclusion.

Indeed the display of temperatures in Fig. 4 is problematic, the challenge being resolution of small temperature differences in the presence of large gradients towards the coast. Therefore, Fig. 4 was removed in the revised manuscript keeping, however, Fig. 4a which now occurs as Fig. 1c. The colour scale was changed in such a way that now features corresponding with the FTLE field in Fig. 1a are clearly recognizable.

5) Finally, and very importantly in my opinion, new diagnostics and metrics should be investigated, related to the observing system as mentioned in point 1). How can LCS be used to evaluate the observing system? How do LCSs vary on time? At which scales? Which proxies can we use to quantify these changes?

The first question (*How do LCSs vary on time?*) can hardly be answered based on a limited number of example situations shown in static figures of the manuscript. This is why a video was provided in the supplement, showing the evolution of FTLE fields over the full year 2016. This video also answers your second question (*At which scales?*). Features at very different scales can be distinguished, arising from an interaction between changing wind conditions and topographic constraints. In the revised manuscript, a summary of wind conditions during the last ten days has been added in each example and also in the supplementary video.

Your third question (*Which proxies can we use to quantify these changes?*) is the most difficult one. It is addressed in the last paragraph of the conclusions. It would be extremely useful if LCS could be predicted via a simple dependence on atmospheric forcing. However, as the LCSs may depend on wind histories over extended periods of time, such a relationship cannot easily be established. The display of recent wind history in each example illustrates the difficulty of establishing a clear relationship.

**More specific points**

**The Introduction (and possibly also the title) should be re-written with more focus to-ward point 1) above.**

The introduction has been revised. Parts of the discussion were moved to the introduction and shortened. A special paragraph was added (starting at line 56), focussing on the use of drift simulations for an improved interpretation of observational data. It is now clearly stated that the study does not aim at the optimization of a monitoring network.

Title: Slightly changed, now explicitly mentioning 'backward simulations'.

More in details, many phrases are unclear. Some examples are listed below - below line 20: "deficiencies of the underlying hydrodynamics...". Is this phrase indicating subgrid uncertainties or what? Deficiencies is certainly not the right word - around line 35.

'Deficiencies' has been replaced by 'inaccuracies'. Indeed these also arise from subgrid uncertainties, as explained by the following half sentence "... including the effects of unresolved sub-grid scale hydrodynamic structures".

**Discussion on local-versus nonlocal is not very precise. Indeed, local relative dispersion has been shown to be much faster at small scales and initial times than non local (Poje et al., 2014).**

I agree that definition of local relative dispersion was missing. This has been added.

**It should also be clarified throughout the text whether the emphasis is on mesoscale or submesoscale dynamics**

In the discussion it is mentioned (starting at line 304) that the scale of ridges in the FTLE field can be smaller than grid resolution in the underlying field, due to the fact that separation of tracer particles released vary close together may evolve along drift paths that much exceed numerical grid resolution. This fact makes it very difficult to classify FTLE ridges in terms of either mesoscale or submesoscale dynamics.

Section 2.1. It would be useful to mention from the beginning (lines 80-85) the ge-ographical extension of the German Bight (lat/long are now mentioned at line 128 in Section 2.4), and clarify that the area is depicted in all the figures.

This information has been added in Section 2.1.

Section 3, on results. The author shows 3 examples of LCS (Fig1, 2) for three different flow realizations and dates, 1 example of particle stats (Fig.3) for another realization, and finally SST (Fig.4) for a mix of realizations. It would be better to focus on 3 cases only, and compare LCS with particle stats, as well as SST.

Thanks for your suggestion. In the revised manuscript, the temperature field in the former Fig. 4a has been moved to Fig. 1 to be combined with the corresponding FTLE and FDLD fields shown in panels (a) and (b). The other panels in former Fig. 4 have been removed. In Fig. 3, a third panel has been added that now complements the analysis by showing the corresponding FTLE field. This FTLE field was previously provided only in the supplement. The FTLE field in Fig. 2a is new, it nicely illustrates the origin of FTLE ridges in Fig. 2b. All changes were done to achieve a better description of the relevance of changing wind conditions.

Section 4, provides a broad discussion on FTLEs and their applications, but there is no clear connection with the present results. Indeed, most of the information are more suitable for the introduction, and in any case should be trimmed and focused on the paper's goals.

Section 4 (Discussion) has been thoroughly revised. A key change the discussion now refers to is that now all examples (figures) include kind of wind rose that summarizes the recent history of winds that lead to the FTLE field shown. The examples suggest that even though strong winds generate the pronounced FTLE ridges, these patterns continue to exist for longer times even under subsequent calm meteorological conditions.

Following the reviewer's advice, some paragraphs were moved from the discussion to the introduction.

---

## Author Comment (AC4)

**I very much appreciate the time and efforts the referee spent on reviewing this paper.**
**In the following, referee comments are given in blue and responses in black.**

*Here, I try to provide my comments about this manuscript.*

*In general, the topic is appropriate for this journal, and I'm not aware of a similar study for the German Bight, which it makes it worth of publishing.*

*The used tools (PELETS in combination with surface current velocities obtained from BSH) are state-of-the-art and appropriate. Furthermore, the different quantitative measures to identify Lagrangian Coherent Structures (LCS) or objects with similar meaning are appropriate.*

*I must admit I'm not an expert for LCS and similar structures. However, reading the article and the theory included was very informative and helpful for me. Furthermore, using more the one quantitative measure is quite illustrative.*

***Major comments and questions:***

*- Tides are very important for this region. Might it be possible to give some idea how and if the FTLE structures will change when tides are not subtracted from the currents. So, what might happen when not using residual currents.*

You are right that tidal currents are very important in the German Bight area. Accordingly, all drift simulations were conducted using the full hydrodynamic fields stored on a 15min basis. Looking at the trajectories in Fig. 1a, for instance, tidal movements can easily be identified. In addition, however, these trajectories show also the more long-term movements related to residual currents. In the paper, the term 'residual currents' is used only in the discussion, tidal currents were never separated from the total flow fields when conducting numerical simulations.

*- It was very illustrative to see how variable the FTLE structures are in time. What would happen, if one considers time averaged FTLE-fields. Perhaps averaged over a season or half a year. Will there be stationary or more robust FTLE ridges visible?*

I wasn't able to find any meaningful non-trivial mean fields. The reason for this is that even when similar structures occur, the sharp FTLE ridges usually reside in different locations, so that any averaging results in extremely smooth, non-informative fields.

Due to the strong influence of wind forcing, a classification with regard to wind directions would probably be the most promising approach. In the revised manuscript, each figure now contains a wind rose that summarizes the recent history of wind conditions. Winds are never constant and the recent history of wind forcing differs for each specific date. This means that FTLE structures can hardly be classified in a simple way. The only chance might be some more fuzzy classification of FTLE fields which, however, could be quite a challenge.

*- Instead of showing the temperature and salinity fields, perhaps one could use the density distribution. Would this do any difference to the interpretation of the relationship between FTLE ridges and T/S fields.*

Temperature and salinity were used just as simple indicators of the accumulated effects of surface divergences over some period of time. Surface temperatures have the advantage that they could even be observed by remote sensing. The problem with finding a proper colour scale was solved in the revised manuscript (see Fig. 1c therein). Density fields could also be interesting to look at, but unfortunately these fields were not be available from the archived data underlying this study.

*Some minor comments and questions:*

*- How do you calculate the residual currents from the BSH data? The current velocities of the BSH include tides, which are important for this region. Are the results dependend on the way these residual currents are calculated?*

All simulations were based on hydrodynamic fields including tidal effects. See my comments above.

*- Do you think it might be helpful to plot the absolute value of the gradient of temperature or salinity instead of the pure fields. So, one could see the sharp fronts more clearly.*

This might indeed be one option to solve the problem of showing small local differences in the presence of substantial changes on a larger scale. I decided, however, to skip the former Fig. 4, keeping just Fig. 4a, which is now presented as Fig. 1c. The colour scale was changed in such a way that now the structure corresponding with the FTLE and FDLD fields in Fig. 1a and 1b can well be recognized. Higher temperatures very near to the coast were discarded in the revised plot.

*Some direct comments to the text:*

*- Line 13: Gap at the end of sentence; (2017). Stanev . . .*

Corrected

*- Line 143: The year of the Huntley citation is missing*

Corrected

*- Equation (5): It is divergence of the 2d surface velocity field, isn't it?*

Yes, thanks for the hint. I changed symbol $\nabla$ into $\nabla_H$ and mentioned 'horizontal divergence' in the text.

*- Figure 2a: The endpoint of the red drifter starting at 54 30' is missing?*

Has been corrected, thanks for checking the plot so carefully!

*- Perhaps one could also include the river Ems to the plots.*

Location of river Ems is now indicated in all figures.

*- Figure 4, caption: . . . times of the FTLE fields . . . ?*

Has been changed.

*- Additional figure S3: The unit of salinity should be different from PSU. Perhaps no unit or g/kg*

Thanks for this advice, but the salinity was removed from the revised paper, focussing on temperature distribution, which is now shown in Fig. 1c.

---

## Author Response (AR1)

**Dear editor,**

in my opinion the revised manuscript gained a lot from the very constructive comments by two referees and the discussants Rodrigo Duran and Jens Meyerjürgens.

Addressing the concerns by Rodrigo Duran, the manuscript now much better deals with the distinction between convergence and divergence-free confluence. The theoretical basis is provided in Section 2.5, in the discussion a new figure (Fig. 6) and Table 1 have been added. Another major improvement is that now each example situation is supplemented with a wind rose that summarizes wind conditions during the trajectories' travel time. This helps to understand the role of strong winds for the generation of FTLE ridges and to identify memory effects during subsequent calm conditions.

Please find below my detailed responses (in black) to the referees' criticism and suggestions (in blue).

**Referee #1:**

First, I would like to thank the referee for the time spent on the thorough review of this paper.

The manuscript investigates Lagrangian metrics such as FTLE and absolute/relative dispersion using model results in the German Bight, showing the high variability of LCSs and enhanced space variability of trajectories close to ridges. The results are interesting but certainly not novel or unexpected, and the paper lacks in my opinion of clear focus and motivation.

I am not aware of any analysis of LCSs in the German Bight area. Results of such studies will always be site specific, therefore I do not agree that the shapes of the LCSs obtained are trivial or were foreseeable. In my opinion, the sometimes very sharp ridges in the FTLE field come as a surprise.

Fig. 3b (former Fig. 2a) provides the example of a situation in which FTLE ridges are obviously constrained by the specific German Bight bathymetry. To make this more clear, bathymetry is now explicitly shown in the new Fig. 1.

The general objective of the study is to make observers aware of the fact that beyond well known random dispersion, there are coherent structures along which the separation of simulated backward trajectories is shaped more systematically. This objective is now stated more clearly in the second paragraph of the introduction (lines 18-27).

**The author mentions several motivating applications, such as characterization or guidance for the observing system, but it is unclear how this would be carry out.**

'Guidance for the observing system' suggests that the distribution of LCSs would be used for a characterization of certain locations. I revised the discussion to make clear that this is not the intention of this study. The idea is rather that detailed simulations addressing a specific period of observations might support an analysis of these data, helping to better distinguish between real changes in a system and just a shift in the observed water bodies' origins. This point is now better addressed in the discussion (e.g. lines 302-312) and also clearly stated in the introduction (lines 58-59).

I think the paper needs an extensive revision or even better a re-submission, where the motivations and the elements of novelty are clearly indicated and developed. Also, there are several specific points that need further clarification, as detailed in the following.

**Main points**

**1) As mentioned above, there is an extensive literature showing the high sensitivity of particle trajectories to their seeding and the use of LCS to characterize it, so the results presented here are not new.**

The reviewer is right, a large body of literature on LCSs already exists. Nevertheless, the results from this study are new in the sense that so far no such study has been conducted for the German Bight area. The present study describes LCSs that may affect a proper interpretation of German Bight monitoring data. I do definitely not agree that the shapes of these large LCSs, partly constrained by the bathymetry in this specific region, are trivial or were foreseeable.

I think the paper needs a novel angle, and a more specific motivation to make the present results new and interesting. I thought that the angle mentioned by the author in the Introduction regarding the characterization of an observing system composed of fixed points is interesting. But it needs more focus and more practical applications.

With regard to the characterization of an observation system, please see my previous response. This study suggests using hydrodynamic simulations in support of data analysis. I hope that this point is now explained more clearly in both introduction (e.g. lines 55-63) and discussion (e.g. lines 307-312). Optimizing the site locations of an observation network would be another interesting task, challenging due to the high variability of LCSs.

The analysis of any specific data set would go far beyond the scope of the present paper. Such a study would have to address many data specific problems, most of them unrelated to marine transports.

For instance, could the results be used to quantify uncertainty at the stations using as proxies the distance from ridges? Or could they be used to indicate areas of influence of the stations, in terms of LCS patterns of dispersion properties? Investigating this type of questions would be very useful from the application point of view and could lead to new results.

I see the reviewer's point. However, in the light of high variability of the LCSs it seems difficult (if not impossible) to define such characteristic areas of influence for specific stations. The areas observations are representative for at a given time much depend on the recent history of winds. Instead of a static characterization of monitoring stations, a model based representation of wind driven, variable hydrodynamic conditions is supposed to be useful to support the interpretation of monitoring data.

In the revised manuscript, all figures now provide a summary of wind conditions over the last 10 days. An interesting result is that pronounced LCSs initiated by strong winds may survive and even sharpen under subsequent calm wind conditions.

2) Since the results are based on the BSH model outputs, it is very important that the model set up and its validation are adequately described, significantly improving Section 2.2. This is especially relevant since the model based results are envisioned to be used in support of the observing system, possibly also in real time.

The description in Section 2.2 has been extended, it now provides more detailed information about the model.

BSHcmod has been in operational use for about two decades, with revisions being applied where needed and possible. Practical experiences were gained also in the context of search and rescue. The operational model was chosen exactly for the reason the referee mentions: BSHcmod results are

reliably available on an everyday basis. Of course, higher spatial resolution would be useful, but for the moment such more expensive simulations are not available operationally.

**More specifically, has Lagrangian validation ever been performed using drifters?**

Yes, the same model setup plus a small leeway (0.6% of 10m winds) has successfully been used for the evaluation of drifter experiments. Corresponsing references (Callies et al. 2017b, 2019) have been added at the end of Section 2.2 (line 119).

**It is also important to be up front regarding model limitations. For instance, given the 1 km resolution we can expect that coastal submesoscale is only partially resolved at best.**

Thanks, this is indeed an important aspect, now addressed in the discussion (lines 326-330): "Computationally more demanding FTLE analyses on a finer grid would have enabled identification of structures even smaller than resolution of the Eulerian hydrodynamic model arising, however, from tracer simulations over longer distances (Huhn et al., 2012). This shows that a classification of kinematic LCSs in terms of mesoscale or submesoscale features and processes may be difficult. Longer integration periods underlying the LCS analysis may filter more short-term features (Serra and Haller, 2016)."

**Also, if the model is hydrostatic, we cannot expect that near surface divergence processes are correctly described.**

The model is hydrostatic, but in the absence of significant bathymetric variations implying large vertical velocities, this shouldn't be a major restriction. The present analysis focusses on open sea conditions, where currents proved to provide a reliable basis for drift simulations (Callies et al., 2017b, 2019).

**3) The description of the used techniques in Section 2.4- 2.5 should be improved, indicating also possible limitations and clarifying definitions. For instance, is the definition of FTLE in eq (3) valid in the case of 2-dimensional flows (as the text at line 120 seems to imply)?**

**The theoretical concepts are now explained giving more details.**

Section 2.4: The formalism deals with a purely kinematic description of flow motions (according to Haller (2015): "...*mimicking experimental flow visualization by tracers"*) so that there are no limitations with regard to the underlying physical dynamics. The whole study deals with two-dimensional surface currents, which is now more obvious from the 2D deformation gradient being explicitly specified in Eq. (2).

**Also, what is the difference between eq (4) and (5) for dilation? From the text (line 144-145), they seem to indicate the same thing, but it is unclear. Indeed the results in Fig.1b,c are quite different.**

The reviewer is right that in the original paper the values in Fig. 1b and Fig.1c differed by a factor of 2. This factor was due to a mistake. Erroneously, dilation rate was calculated from the eigenvalues of the Cauchy-Green strain tensor rather than the singular values of the deformation tensor. As these eigenvalues equal the squared singular values (line 146 in the revised manuscript), this error resulted in a factor of two when the logarithm was taken.

Having corrected this error, the FDLD and dilation rate differ just due to different numerical discretization. While the dilation rate is calculated from the deformation gradient defined on the 1km FTLE grid, calculation of the Eulerian divergences uses auxiliary points at a 250m distance (see explanation in lines 184-186).

Because FDLE and dilation rate are so similar, I removed dilation rate from the figure (now Fig. 2) and replaced it by the temperature plot (formerly Fig. 4a, now Fig. 2c). The colour maps of both Fig. 2b and Fig. 2c have been modified to make related structures better visible.

4) In general, I think that the text commenting the results should be more realistic throughout the paper. For instance the comparison between LCS ridges, and the 2 forms of dilation in Fig.1 (lines 188-195 and lines 312-13) is very positive, while I fail to see a good comparison between the figures. I do not see a "striking similarity" between Fig.1a and 1b, where the main North-South ridge is absent.

I do not understand why the reviewer claims that he does not see the North-South ridge in Fig. 1b (now Fig. 2b). Actually, the FTLE ridge from Fig. 1a (now Fig. 2a) is reproduced very clearly in terms of a line of convergence. To improve visibility, the colour scale of Fig. 1b (now Fig. 2b) has been changed, replacing green by brown.

**The author acknowledges the clear difference between Fig.1b and 1c, but I do not understand the point of the comparison, given that the model itself is not well suited for this diagnostics.**

Why should the model not be suited? I guess that the reviewer refers to the factor of 2, which (see my above explanation) was an error that has been corrected. Nevertheless, Fig. 1c (now Fig. 2c) showing the dilation rate has been removed and replaced by the temperature distribution (formerly Fig. 4a, see above). In this way, all relevant information for the particular example (12 June 2015) is now concentrated in one figure and can more easily be assessed.

Also the comments on Fig.4 do not seem very grounded to me. In a case with very little gradients, except for the obvious coastal ones, as in Fig.4a and at some extent 4c, it is impossible to draw any meaningful conclusion.

I agree with the reviewer that the display of temperatures in Fig. 4 is problematic indeed, the challenge being the resolution of small temperature differences in the presence of large gradients towards the coast. Therefore, in the revise manuscript Fig. 4 was removed, keeping only Fig. 4a which now occurs as Fig. 2c. The colour scale was changed in such a way that now features corresponding with the FTLE field in Fig. 2a should be well recognizable.

**5) Finally, and very importantly in my opinion, new diagnostics and metrics should be investigated, related to the observing system as mentioned in point 1). How can LCS be used to evaluate the observing system? How do LCSs vary on time? At which scales? Which proxies can we use to quantify these changes?**

The first question (*How do LCSs vary on time?*) can hardly be answered based on a limited number of example situations shown in static figures of the manuscript. This is why a video was provided in the supplement, showing the evolution of FTLE fields over the full year 2016. This video also answers your second question (*At which scales?*). Features at very different scales can be distinguished, arising from an interaction between variable wind conditions and topographic constraints. In the revised manuscript, a summary of wind conditions during the last ten days has been added in each example and also in the supplementary video.

The referee's third question (*Which proxies can we use to quantify these changes?*) is the most difficult one. It is addressed in the last paragraph of the conclusions (lines 414-419). It would be very useful if LCS could be predicted as a simple function of atmospheric forcing. However, the LCSs depend on detailed past wind histories over extended periods of time, so that such relationship cannot easily be established. The display of recent wind history in each example illustrates the difficulty.

**More specific points**

**The Introduction (and possibly also the title) should be re-written with more focus to-ward point 1) above.**

The introduction has been revised. Parts of the discussion were moved to the introduction and shortened. A special paragraph was added (starting at line 55), focussing on the use of drift simulations for an improved interpretation of observational data. It is now clearly stated that the study does not aim at the optimization of a monitoring network.

Title: Slightly changed, now explicitly mentioning 'backward simulations'.

More in details, many phrases are unclear. Some examples are listed below - below line 20: "deficiencies of the underlying hydrodynamics...". Is this phrase indicating subgrid uncertainties or what? Deficiencies is certainly not the right word - around line 35.

I agree, the whole sentence has been deleted.

Discussion on local-versus nonlocal is not very precise. Indeed, local relative dispersion has been shown to be much faster at small scales and initial times than non local (Poje et al., 2014).

I agree that a definition of local relative dispersion was missing. This has been added.

**It should also be clarified throughout the text whether the emphasis is on mesoscale or submesoscale dynamics**

In the discussion it is now mentioned (lines 326-330) that the scale of ridges in the FTLE field can be smaller than grid resolution in the underlying field. Separation of tracer particles released very close together may evolve along drift paths that much exceed numerical grid resolution. This fact makes it very difficult to classify FTLE ridges in terms of either mesoscale or submesoscale dynamics. These ridges provide a merely kinematic description.

Section 2.1. It would be useful to mention from the beginning (lines 80-85) the geographical extension of the German Bight (lat/long are now mentioned at line 128 in Section 2.4), and clarify that the area is depicted in all the figures.

This information has been added in Section 2.1 (line 88).

Section 3, on results. The author shows 3 examples of LCS (Fig1, 2) for three different flow realizations and dates, 1 example of particle stats (Fig.3) for another realization, and finally SST (Fig.4) for a mix of realizations. It would be better to focus on 3 cases only, and compare LCS with particle stats, as well as SST.

Thanks for this suggestion. In the revised manuscript, the temperature field in the former Fig. 4a has been moved to Fig. 2 to be combined with the corresponding FTLE and FDLD fields shown in panels (a) and (b). The two other panels in former Fig. 4 have been removed, together with section 3.2 on surface temperatures. To complement the analysis, in Fig. 3 (now Fig. 4), a third panel has been added, showing the corresponding FTLE field (previously provided only in the supplement). The FTLE field in Fig. 3a is new, it nicely illustrates the origin of FTLE ridges in Fig. 3b about 8 days later, demonstrating the relevance of changing wind conditions.

Section 4, provides a broad discussion on FTLEs and their applications, but there is no clear connection with the present results. Indeed, most of the information are more suitable for the introduction, and in any case should be trimmed and focused on the paper's goals.

Section 4 (Discussion) has been thoroughly revised. Following the reviewer's advice, some paragraphs were moved from the discussion to the introduction. A new key issue in the discussion (raised by Rodrigo Duran) is the question to which extent FTLE ridges may be seen as lines of convergence rather than convergence-free confluence (lines 331-363). In the context of this discussion, Fig. 6 and Table 1 have been added. Another important aspect being addressed is the role of strong winds for the generation of FTLE ridges and memory effects during subsequent calm conditions (lines 313-325). The problem of dealing with possible simulation errors is addressed at lines 377-385.

**Referee #2:**

Once more I would like to thank the referee for the time spent on reviewing my paper.

Here, I try to provide my comments about this manuscript.

In general, the topic is appropriate for this journal, and I'm not aware of a similar study for the German Bight, which it makes it worth of publishing.

The used tools (PELETS in combination with surface current velocities obtained from BSH) are stateof-the-art and appropriate. Furthermore, the different quantitative measures to identify Lagrangian Coherent Structures (LCS) or objects with similar meaning are appropriate.

I must admit I'm not an expert for LCS and similar structures. However, reading the article and the theory included was very informative and helpful for me. Furthermore, using more the one quantitative measure is quite illustrative.

**Major comments and questions:**

- Tides are very important for this region. Might it be possible to give some idea how and if the FTLE structures will change when tides are not subtracted from the currents. So, what might happen when not using residual currents.

The referee is right that tidal currents are very important in the German Bight area. Accordingly, all drift simulations were conducted using the full hydrodynamic fields stored on a 15min basis. Looking at the trajectories in Fig. 2a (revised manuscript), for instance, tidal movements can easily be identified. In addition, these trajectories show the more long-term movements related to residual currents. However, tidal currents were never separated from the total flow fields when conducting numerical simulations.

**- It was very illustrative to see how variable the FTLE structures are in time. What would happen, if one considers time averaged FTLE-fields. Perhaps averaged over a season or half a year. Will there be stationary or more robust FTLE ridges visible?**

I wasn't able to find any meaningful non-trivial mean fields. The reason for this is that even when similar structures occur, the sharp FTLE ridges usually reside in different locations, so that any averaging results in extremely smooth, non-informative fields.

Due to the strong influence of wind forcing, a classification with regard to wind directions would probably be the most promising approach. In the revised manuscript, each figure now contains a wind rose that summarizes the recent history of wind conditions. Winds are never constant and the recent history of wind forcing differs for each specific date. This means that resulting FTLE structures can hardly be classified in a simple way. The only chance might be some more fuzzy classification of FTLE fields.

**- Instead of showing the temperature and salinity fields, perhaps one could use the density distribution. Would this do any difference to the interpretation of the relationship between FTLE ridges and T/S fields.**

Temperature and salinity were used just as simple indicators of the accumulated effects of surface divergences over some period of time. In the revised manuscript, the whole section on surface temperatures has been removed together with the temperature fields in Fig. 4. Just panel Fig. 4a was kept, being shown as Fig. 2c in the revised manuscript. The problem with finding a proper colour map could be solved after the exclusion of very nearshore regions. Salinity fields are no longer considered.

Surface temperatures have the advantage that they can be observed by remote sensing. Density fields would be interesting to look at, but unfortunately these fields were not available from the archived data underlying this study.

**Some minor comments and questions:**

- How do you calculate the residual currents from the BSH data? The current velocities of the BSH include tides, which are important for this region. Are the results dependend on the way these residual currents are calculated?

All simulations were based on hydrodynamic fields including tidal effects. See my comments above.

- Do you think it might be helpful to plot the absolute value of the gradient of temperature or salinity instead of the pure fields. So, one could see the sharp fronts more clearly.

This might indeed be one option to solve the problem of showing small local differences in the presence of substantial changes on a larger scale. I decided, however, to remove the former Fig. 4, keeping just Fig. 4a, which is now presented as Fig. 2c. The colour scale was changed in such a way that now the structure corresponding with the FTLE and FDLD fields in Fig. 2a and 2b can well be recognized.

**Some direct comments to the text:**

- Line 13: Gap at the end of sentence; (2017). Stanev . . .

Corrected

- Line 143: The year of the Huntley citation is missing

Corrected

- Equation (5): It is divergence of the 2d surface velocity field, isn't it?

Yes, thanks for the hint. I changed symbol  $\nabla$  into  $\nabla_H$  and mentioned 'horizontal divergence' in the text.

- Figure 2a: The endpoint of the red drifter starting at 54 30' is missing?

Has been corrected, thanks for checking the plot so carefully!

- Perhaps one could also include the river Ems to the plots.

Location of river Ems is now indicated in all figures.

- Figure 4, caption: . . . times of the FTLE fields . . . ?

Has been changed.

- Additional figure S3: The unit of salinity should be different from PSU. Perhaps no unit or g/kg

Thanks for the advice, but the consideration of salinity was removed from the paper.

---

## Author Response (AR2)

Dear Editor,

please find below my responses (in black) to the referee's comments (in blue). Again I would like to thank the referee for his efforts and constructive remarks.

I would also like to mention that I slightly changed notation in Eqs. (10) and (11), explicitly adding the list of arguments. This makes notation more consistent with the notation used in Eq. (5) but does not affect the content or meaning of the equations. I also added the reference to Staneva et al. (2021), a relevant paper published in these days.

I found the manuscript significantly improved. The objectives of the paper have been clarified and are now more realistic, the descriptions of the model and methodology have been expanded, and the description of results and associated figures are better organized and clearer.

I think that my main concerns have been addressed, and I recommend the paper for publication.

There are still some minor points listed in the following that I would like the author to consider before publication, but I do not need to see the final revision (lines in the following refer to the edited version):

- Line 10: The last phrase of the Abstract is not very clear, especially the word "vagueness" does not seem correct: "indicating also vagueness of drift simulations being used." I recommend to delete the phrase or to change it with someting like "indicating that direct backward simulations of trajectories can be very sensitive and therefore unreliable"?

Has been changed to: "... indicating when simulations of backward trajectories are unreliable because of their high sensitivity to tracer seeding positions."

- Line 54: Typo: "weakly" instead of "weekly"

Thanks, has been corrected.

- Line 90: phrase not clear, may be a verb is missing?

The wording has been revised.

- Line 110: "provides" instead of "tries to give"

Has been replaced.

- Section 2: I am still confused about using a hydrostatic model to provide estimates of surface convergence. Don't you expect that vertical velocity estimates are not very reliable? Please add a comment.

The hydrostatic approximation refers to vertical accelerations rather than vertical velocities, assuming that these accelerations are substantially smaller than gravitational acceleration. Given the Boussinesq approximation, vertical velocities are obtained from the constraint of three-dimensional divergences being equal to zero. There shouldn't be any problem with these assumptions, in particular when the relevant horizontal scales are much larger than the vertical scales. This is the case for the shallow shelf sea (see Fig. 1). Note also that in this study vertical velocities are never used or addressed explicitly. The FDLD (Eq. 11) is

calculated from 2D horizontal velocities. Therefore, I couldn't find a place in the manuscript where it would be meaningful to discuss this issue.

- Section 3: I think the presentation is much improved, but there is still some confusion. I think that there should be a brief discussion at the beginning of the Section that explains the rational behind the choice of examples. Are they driven by the type of FTLE structure? Or by the type of wind forcing? Or what else? Also the titles of the subsections "First example" and so on are a bit missleading, in the sense that in some cases there are several examples grouped together... It might be better to refer to the type of example, as indicated above

Yes, I see your point. The very generic titles of subsections have now been replaced by specific titles that summarize the main issue addressed in the respective subsection.

- **Discussion on the choice of tau=250 h**: I think that motivation, rational and sensitivity of this choice should be put up front in Section 3, not at the end and in the Discussion

In agreement with your suggestion, a motivating statement has been inserted at the beginning of Section 3.